# Modelling sun-induced fluorescence and photosynthesis with a land surface model at local and regional scales in northern Europe

Tea Thum[1], Sönke Zaehle[2], Philipp Köhler[3], Tuula Aalto[1], Mika Aurela[4], Luis Guanter[3], Pasi Kolari[5], Tuomas Laurila[4],
Annalea Lohila[4], Federico Magnani[6], Christiaan Van Der Tol[7],Tiina Markkanen[1]

[1]Climate Research, Finnish Meteorological Institute, P.O. Box 503, 00101 Helsinki, Finland
[2]Biochemical Integration Department, Max Planck Institute for Biogeochemistry, Hans-Knöll-Str. 10, 07745 Jena, Germany
[3]Helmholtz-Centre Potsdam, GFZ German Research Centre for Geosciences, Section 1.4 Remote Sensing, Telegrafenberg, 14473 Potsdam, Germany
[4]Atmospheric Composition Research, Finnish Meteorological Institute, P.O. Box 503, 00101 Helsinki, Finland
[5]Department of Physics, University of Helsinki, 00014 University of Helsinki, Finland
[6]University of Bologna, Via Zamboni, 33, 40126 Bologna, Italy
[7]Deparment of Water Resources, Faculty ITC, University of Twente, P.O. Box 217, 7500 AE Enschede, Netherlands

*Correspondence to*: Tea Thum (tea.thum@fmi.fi)

**Abstract.** Recent satellite observations of sun-induced chlorophyll fluorescence (SIF) are thought to provide a large-scale proxy for gross primary production (GPP), thus providing a new way to assess the performance of land surface models (LSMs). In this study, we assessed how well SIF is able to predict GPP in the Fenno-Scandinavian region and what potential limitations for its application exist. We implemented a SIF model into the JSBACH LSM and used active leaf level chlorophyll fluorescence measurements (ChlF) to evaluate the performance of the SIF module at a coniferous forest at Hyytiälä, Finland. We also compared simulated GPP and SIF at four Finnish micrometeorological flux measurement sites to observed GPP as well as to satellite observed SIF. Finally, we conducted a regional model simulation for the Fenno-Scandinavian region with JSBACH and compared the results to SIF retrievals from the GOME-2 (Global Ozone Monitoring Experiment-2) space-borne spectrometer and to observation-based regional GPP estimates. Both observations and simulations revealed that SIF can be used to estimate GPP at both site and regional scales. At regional scale the model was able to simulate observed SIF averaged over five years with $r^2$ of 0.86. The GOME-2 based SIF was a better proxy for GPP than the remotely sensed fAPAR (fraction of absorbed photosynthetic active radiation by vegetation).The observed SIF captured the seasonality of the photosynthesis at site scale and showed feasibility for use in improving of model seasonality at site and regional scale.

**List of abbreviations**

$\Phi_f$ - Quantum yield of fluorescence [quanta emitted / quanta absorbed]

$\Phi_{f,s}$ – Scaled quantum yield of fluorescence [quanta emitted / quanta absorbed]

$\Phi_p$ – Quantum yield of photochemistry in PSII [electrons transported / quanta absorbed]

ChlF – chlorophyll fluorescence (general term)

ESM – Earth System Model

EVI – Enhanced Vegetation Index

F' – Prevailing fluorescence signal as measured with PAM fluorometry [relative units, e.g. sensor mV output]

fAPAR – Fraction of absorbed photosynthetic active radiation by vegetation

FTS - Fourier Transform Spectrometer; spectrometer on GOSAT satellite

GOME-2 – Global Ozone Monitoring Experiment-2; spectrometer on MetOp-A (Meteorological Operational Satellites) satellite

GOSAT – Greenhouse Gases Observing Satellite

GPP – Gross Primary Production

JSBACH – Jena Scheme for Biosphere Atmosphere Coupling in Hamburg; the land surface model of the Max Planck Institute's Earth System

Model

LSM – Land Surface Model

NDVI - Normalized Difference Vegetation Index

PAR – Photosynthetically active radiation

PSII – Photosystem II

SCIAMACHY - SCanning Imaging Absorption SpectroMeter for Atmospheric CHartographY; spectrometer on ENVISAT (ENVIronmental

SATellite) satellite

SCOPE - Soil-Canopy Observation of Photosynthesis and Energy

SIF – Sun-induced fluorescence [e.g. in W m$^{-2}$ sr$^{-1}$ nm$^{-1}$], obtained from passive observations or from a model

## 1 Intoduction

The terrestrial biosphere is thought to store approximately a quarter of the carbon dioxide ($CO_2$) released by anthropogenic activity (Le Quéré et al., 2016). However, a detailed spatio-temporal distribution of this uptake is absent, partly due to an incomplete understanding of the terrestrial carbon balance as a whole. Estimates of the terrestrial net carbon balance are often made by land surface models (LSMs) (Sitch et al., 2015). However, assessing and improving the performance of LSMs at larger scales remains a challenge, as limited data sources for large scale carbon dioxide flux estimates are available (Luo et al., 2012). Increasing our knowledge of carbon dioxide uptake will thus help to provide better estimates of the global carbon balance.

Previous global estimates of the spatial distribution and the variability of plant photosynthetic production have mostly been based on remote sensing of vegetation greenness (such as the normalized difference vegetation index, NDVI) or the fraction of absorbed radiation (fAPAR) describing how much of the incoming photosynthetically active radiation (PAR) is absorbed by the vegetation (Pinty et al., 2011). Recently, global retrievals of sun-induced fluorescence (SIF) have also become available for the monitoring of global vegetation productivity (e.g. Frankenberg et al., 2011, Joiner et al., 2011).

Chlorophyll fluorescence (ChlF) takes place in plant leaves when they photosynthesize. The light energy absorbed by the chlorophyll molecules is used in photosynthesis, dissipated as heat or re-emitted as light through ChlF (Maxwell and Johnson, 2000). Thus, ChlF correlates with two simultaneous processes: photosynthesis and heat dissipation. Therefore, ChlF has been a standard measurement at the leaf scale in plant physiology for decades (Baker, 2008). The advent of retrieval approaches for satellite data acquired by the spectrometers FTS (onboard satellite GOSAT), SCIAMACHY (satellite ENVISAT), GOME-2 (satellite MetOp-A and B) and OCO-2 have demonstrated that it is possible to measure SIF from space (e.g. Frankenberg et al., 2011, 2014; Guanter et al., 2012, Joiner et al., 2012, 2013; Köhler et al. 2015).

As the seasonal cycles of fAPAR and SIF are related to radiation and greenness of the vegetation, they appear similar in many ecosystems. However, SIF is more physiologically related to photosynthesis and has been shown to track gross primary production (GPP) better than fAPAR in deciduous broadleaf and mixed forests as well as in croplands (Joiner et al., 2014). Moreover, comparison to observation-based upscaled global GPP products (Jung et al., 2011) has suggested that SIF is a better estimator of GPP than other traditional remotely sensed vegetation indices, such as EVI (enhanced vegetation index) and NDVI (Frankenberg et al., 2011; Walther et al., 2016), and may thus be of relevance in the observation or modelling of the terrestrial carbon balance (Lee et al., 2015; Parazoo et al., 2013).

SIF can be estimated from state-of-the-art photosynthesis models, such as the widely used Farquhar model (Farquhar et al., 1980), by describing the processes of photosynthesis and fluorescence at the cellular level (van der Tol et al., 2009a), or leaf level (van der Tol et al., 2014). The strong dependence of the measurable SIF signal on scattering and reabsorption effects within the canopy requires explicit formulation of the radiative transfer (e.g. SCOPE; van der Tol et al. 2009b). Nevertheless, modelling studies using satellite observed SIF have revealed links between forest

productivity and water stress in the Amazon (Lee et al., 2013) and have helped to constrain the seasonal cycle of GPP (Parazoo et al., 2014). Koffi et al. (2015) included SIF in their global carbon cycle data assimilation system and found SIF to be more sensitive to the chlorophyll content in the leaves than to the parameter maximum carboxylation rate controlling model GPP.

The challenge in using the space-borne SIF data for the evaluation of SIF models is the lack of similar ground-based observations, and the degree

of correspondence between GPP and SIF at different spatial scales. Another important consideration is that the SIF observation from space is dependent on a passive measurement carried out in narrow spectral bands. The signal in the red region originates mostly from the top of the canopy, whereas deeper canopy layers also contribute to the far-red signal (Porcar-Castell et al., 2014). Both these regions can be used in retrieving SIF from remote sensing observations.

Our aim in this study is to assess whether the SIF measurements can be used to quantify the LSM performance at a regional scale for the spring and autumn transition periods. Forests in the boreal zone experience strong seasonal cycle with cold winters and warm summers (Bonan, 2008). The transition periods of spring and autumn influence the carbon balances in these northern ecosystems (Bergh et al., 1998). In changing climate the conditions in spring and autumn will change (Ruosteenoja et al., 2011) and cause changes to carbon balances. It is anyhow during these times that the carbon cycle models have difficulties in performance (Schaefer et al., 2012). Therefore it is important to find ways to improve carbon

cycle models in these time periods.

In Fenno-Scandinavia coniferous forests are very common and study of their photosynthetically active period with remote sensing products is a challenge because of the stronger relative contribution to the GPP cycle of the physiological seasonal cycle than the changes in green foliage area (Böttcher et al., 2014). Our strategy consists of implementing a ChlF model into LSM JSBACH model and evaluating the results of this

implementation. We examined the performance of the model by comparing it to leaf-level ChlF observations at the site scale in one forest. We then evaluated the model performance at four coniferous forest sites by comparing the remotely sensed SIF signal from GOME-2, modelled SIF and the modelled GPP to observations with the eddy covariance technique. Finally, we made a regional model run for Fenno-Scandinavia and compared our results to satellite observations.

## 2 Materials and methods

### 2.1  Active and passive measurements of ChlF in general

Active measurements of ChlF in field conditions are typically done with the pulse amplitude modulated (PAM) technique where ChlF is measured over a broad spectral region (Porcar-Castell et al., 2014). In the active measurement, a weak and pulsed measuring light is used to excite fluorescence. The ChlF measured by PAM is not dependent on the prevailing light environment, but reflects the efficiency in transforming the measuring light into fluorescence. The active measurement provides the fluorescence signal F' and the photosynthesis yield $\Phi_p$, which describes the fraction of absorbed photons used in photosynthesis. For the separation between non-photochemical quenching (NPQ) (i.e.heat dissipation) and $\Phi_p$ it is also necessary to have observations of dark-adapted leaf, as it is assumed that dark-adapted leaf with all the reaction centers open do not exhibit NPQ (Murchie and Lawson, 2013).

Passive measurements are an alternative to active observation of ChlF and rely on the emission under natural light environments, where the SIF is estimated in very narrow spectral bands and is affected by ambient illumination (Porcar-Castell et al., 2014). Passive measurements can be ground-based, but also based on remote sensing carried out on the ground or from space (Meroni et al., 2009). Passive observations rely on the in-filling of atmospheric or solar absorption lines by SIF. The ChlF yield ($\Phi_f$) can be obtained from the passive measurements and it is an indication of the fraction of electrons in the leaf follow the ChlF pathway.

Thus, passive measurements provide SIF and fluorescence yield $\Phi_f$ values, whereas the active measurements provide the prevailing fluorescence signal F' and yield of photosynthesis $\Phi_p$. In non-stressed low light conditions most of the absorbed energy is used for photosynthesis (causing higher $\Phi_p$) that results in lower fluorescence yield ($\Phi_f$). Therefore an inverted relationship exists between ChlF and photosynthesis yields at low light levels (van der Tol et al., 2009a). However, during high light and/or stressed conditions NPQ is increased and then $\Phi_p$ and $\Phi_f$ are positively correlated.

### 2.2 Models

#### 2.2.1 JSBACH

We used the biosphere model JSBACH (Reick et al., 2013) that is part of the Max Planck Institute's Earth System Model (Giorgetta et al., 2013). In addition to the global simulations, JSBACH can also be applied at regional and site scales.

The JSBACH model simulates the exchanges of carbon, water and energy between the land surface and the atmosphere. The incoming radiation that reaches the canopy is calculated for the three different canopy layers a using two-stream approximation model (Dickinson 1983, Sellers 1985). In this model, it is assumed that the distribution of scattering objects in the canopy is homogenous so that the radiation distribution inside the canopy is horizontally invariant. Therefore, it is necessary to only consider vertical radiant fluxes.

Fraction of absorbed PAR (fAPAR) for one layer is calculated as

$$fAPAR(l_n, l_{n+1}) = \frac{I_{tot}(l_n) - I_{tot}(l_{n+1})}{R_{dir}(0) + R_{diff}(0)} \qquad (1)$$

where $l_n$ is the cumulative leaf area index (LAI) for the canopy layer, $I_{tot}$ is the total incoming radiation that reaches the canopy layer and includes the direct incoming radiation to the canopy layer and upward and downward diffuse radiation. $R_{dir}(0)$ is the direct radiation at the top of the canopy and $R_{diff}(0)$ is the incoming diffuse radiation at the top of the canopy. The absorbed radiation for each canopy layer is used in the photosynthesis calculation. The fAPAR for the whole canopy is obtained by summing the values from the three different layers together.

In this model, photosynthesis is described by the Farquhar *et al.* (1980) formulation for C3 plants and stomatal conductance is based on Knorr (2000) (photosynthesis for C4 plants follows Collatz et al. (1992) but these species are not relevant in our study region). Photosynthesis is either electron transport rate or maximum carboxylation rate limited. The electron transport rate *J* from the photosynthesis model is used in the calculation of chlorophyll fluorescence and its formulation is as follows:

$$J(I) = J_{max} \frac{\alpha I}{\sqrt{J_{max}^2 + \alpha^2 I^2}} \qquad (2)$$

where $J_{max}$ is the maximum electron transport rate (unit: $\mu mol\ m^{-2}\ s^{-1}$) with a linear air temperature dependency, $I$ is incoming photosynthetically active radiation (unit: $\mu mol\ m^{-2}\ s^{-1}$) and $\alpha$ is the apparent quantum yield (value 0.28).

The vegetation in JSBACH is described by Plant Functional Types (PFTs). Different PFTs have specific physiological properties, such as photosynthetic capacity and physical properties, such as the albedo of the canopy. Each grid cell can contain up to four different PFTs and we used 13 potentially different PFTs for vegetation in our simulation. The vegetation map was based on the European Corine Database, described in

Törmä et al. (2015). The leaf area development in JSBACH is based on the LoGro-P (Logistic Growth Phenology) model (Böttcher et al., 2016). Air temperature is the main driver of the phenological development in the two main vegetation types (evergreen needleleaf forests and temperate deciduous broadleaf forests) in our study region.

### 2.2.2 Leaf-level chlorophyll fluorescence

The model equations for the leaf-level fluorescence are shown in Appendix A. The outputs of the model are SIF and scaled fluorescence yield $\Phi_{f,s}$. Due to simplifying assumptions such as lack of wavelength separation we do not simulate the magnitude of SIF with JSBACH. However, seasonal changes in the modelled SIF are still captured. In the ChlF related literature, the ChlF quantities are often referred to as parameters, but in this work the word parameter will refer to a model parameter that is kept constant and ChlF related observations are instead referred to as variables.

### 2.2.3 Canopy scaling

In order to derive a comparatively simple, computationally efficient scheme for the emission and extinction of radiation in the fluorescence wavelengths, we developed a simplified parameterization with the form

$$SIF_{can} = \sum_{i=1}^{n_{layers}} SIF_i \cdot e^{\left(-k_{fl}\frac{LAI_{tot}}{n_{layers}} \cdot i\right)} \qquad (3)$$

where $SIF_{can}$ refers to the SIF signal that originates from the whole canopy, $n_{layers}$ is the number of canopy layers in JSBACH, $k_{fl}$ is the attenuation coefficient, $LAI_{tot}$ is the total LAI of the canopy. This equation is based on the output of the comprehensive radiative transfer model SCOPE and describes the radiative transfer, photosynthesis, chlorophyll fluorescence, temperature and energy balance at site-level for a given canopy structure (van der Tol et al., 2009b).

We used the SCOPE model version 1.52b in our study. We derived the parameterization of equation (3) by first calculating the hemispherically integrated top of the canopy value for SIF when emission was coming from only one canopy layer at a time. Thus we obtained a profile of how much of the emission that originated from each canopy layer reached the top of the canopy. Dividing this profile by the emission of the different layers yielded the attenuation of the ChlF signal in the canopy. The derived attenuation coefficient $k_{fl}$ was slightly sensitive to the wavelength considered, while changing the amount of foliar mass/area did not affect the attenuation coefficient. The attenuation coefficient $k_{fl}$ also varied with

the solar hourly angle: $k_{fl}$ for wavelength 740 nm was 0.350 at noon and 0.347 at 10:30 am, which corresponds to the approximate local solar time of the GOME-2 observation. Therefore, we used the attenuation coefficient $k_{fl}$ value 0.347 in our analysis.

## 2.3 Measurements

### 2.3.1 Site level ChlF and carbon dioxide ($CO_2$) flux measurements at Hyytiälä

The ChlF site level measurements are from a Scots pine (*Pinus sylvestris*) forest at Hyytiälä, Finland (61°51ʹN, 24°17ʹE, 180 m a.s.l.) (Kolari et al., 2009). The forest was planted in 1962 after burning and mechanical soil preparation. The soil is a Haplic Podzol on glacial till and the site is of medium fertility (Kolari et al., 2009). The forest also has a sparse understory of Norway spruce (*Picea abies*). The total leaf area index (LAI) is 6 $m^2$ $m^{-2}$ for the Scots pine. The $CO_2$ flux between the vegetation and the atmosphere was measured continuously with a closed-path eddy covariance system that is described in more detail in Rannik et al. (2004) and Mammarella et al. (2009). Gapfilling and flux partitioning are
described in Kolari et al. (2009).

ChlF was measured in the Scots pine needles with a MONITORING-PAM Multi-Channel Chlorophyll Fluorometer (Walz, Effeltrich, Germany) (Porcar-Castell et al., 2008; Porcar-Castell, 2011). The measurement period was 15.8.2008 – 14.8.2009. The measurement system MONI-PAM uses a modulated blue LED light to measure the fluorescence emitted from the leaf sample (Porcar-Castell, 2011). We used the results from an
emitter-detector unit that measured three or four pairs of needles arranged in a leaf clip. The unit was located in the mid-canopy.

The instrument recorded instantaneous fluorescence (F'), maximal fluorescence (Fm') and incident PAR radiation. During nighttime, the maximal fluorescence (Fm) and the minimal fluorescence ($F_o$) were measured. The observations were done every 10 minutes during summer and every 30 minutes during winter. The temperature sensitivity of the LED measuring light was corrected in the absolute fluorescence levels (Porcar-Castell,
2011). From the measured ChlF variables it was possible to calculate the quantum yield of PSII by $\Phi_p = (Fm' – F') / Fm'$.

### 2.3.2 Other $CO_2$ flux measurement sites

In addition to Hyytiälä (FI-Hyy), we used measurements from three other Finnish flux measurement sites. Together these four sites cover a wide latitudinal range, with the two southern sites; Hyytiälä and Kalevansuo(FI-Kns) located in the southern boreal zone. Two sites are located north of the Arctic Circle; Sodankylä (FI-Sod) and Kenttärova (FI-Ken) in the northern boreal zone. FI-Ken is a Norway spruce forest, whereas the other
sites are Scots pine forests. More site information can be found in Table 1.

### 2.3.3 Observations from space, SIF and fAPAR

To obtain estimates for SIF, we used data from GOME-2 (Global Ozone Monitoring Instrument 2), which is an operational medium resolution nadir-viewing UV/visible and near-infrared cross-track scanning spectrometer on-board EUMETSAT's polar orbiting MetOp-A and B (Meteorological Operational Satellites) (Munro et al., 2006). The spectrometer measures the Earth's backscattered radiance and the extraterrestrial solar irradiance. The overpass time of the satellite is around 9:30 am local solar time at the equator, while one revolution takes 100 minutes. Here, we use the GOME-2 SIF data set derived with the approach presented by Köhler et al. (2015). The retrieved SIF data were available for 2007-2011, with an 8-day time resolution and a spatial resolution of 0.5° x 0.5°. Typical SIF error estimates range up to 0.5 mW m$^{-2}$ sr$^{-1}$ nm$^{-1}$ (Köhler et al., 2015). Negative values of observed SIF were removed from the analysis. This will likely introduce a positive bias in the values, but it did not affect the seasonal variations in the observations and thus the quantitative analysis presented here.

In our analysis, we also used the space-observed variable fraction of absorbed photosynthetically active radiation (fAPAR). These values were obtained by partitioning the solar radiation fluxes that were based on inversion of the MODIS broadband white sky surface albedos (Pinty et al., 2011). Temporal resolution was 16-days and spatial resolution one kilometer. Monthly values of fAPAR were used in the analysis.

### 2.4 Simulations

### 2.4.1 Site level simulations

At the site-level, JSBACH was run with observed half-hourly meteorology data (air temperature, shortwave and longwave radiation, specific humidity, wind speed, precipitation) for each site, and the vegetation at the site was prescribed to be an evergreen coniferous forest. ERA-Interim data (Dee et al., 2011) was used to fill the missing values in the meteorological time series. The seasonal maximum of LAI of the model over several years was matched to the observed value. The maximum carboxylation rate $V_{c(max)}$ and maximum potential electron transport rate $J_{max}$ parameters were adjusted so that the modelled GPP matched the magnitude of the observation-based GPP. The two parameters have a fixed ratio and $J_{max}$ was fixed accordingly. No rigorous parameter inversion methods were used, as we did not use the absolute GPP values in our study, but focused more on the seasonal behavior. In JSBACH, these two Farquhar model parameters have a vertical profile, which were here assumed to correspond to the observed vertical distribution of foliar nitrogen content (in units leaf mass per area) at FI-Hyy (Palmroth and Hari, 2001).

A comparison of leaf-scale observations with PAM to site-scale simulated values is somewhat difficult, as the PAM measurement provides the photosynthesis yield $\Phi_p$ and fluorescence signal F', whereas JSBACH includes the "passive" fluorescence yield $\Phi_f$ and when multiplied by the

radiation provides an estimate of SIF. Notwithstanding these differences, the seasonal cycle of both values may be compared in relative terms as both are connected to the activity of the photosynthetic apparatus in the plants. The observed GPP is obtained from the flux tower and is, therefore, at the same scale as the simulation output.

At four micrometeorological measurement sites, the modelled SIF and GPP were compared to observed GPP and satellite observations of SIF. Averages of the 2° x 2° spatial resolution pixels closest to the flux tower from the satellite observations were used with a 20-day time period. Averages were used rather than the closest pixel to the site, as spatial averaging reduces the retrieval data error. At the northern sites some satellite observations from mid-winter were absent due to cloud contamination. Therefore those time periods were omitted from the analysis. In addition, the time period 9–11 am was taken from the model results and flux measurements to allow for comparability with the satellite observations.

### 2.4.2 Regional scale simulations

The modelling domain consisted of Fenno-Scandinavia (52° – 74°N, 4° – 44°E). The meteorological data for JSBACH was prepared with the regional climate model REMO (Jacob and Podzun 1997, Jacob 2001, Jacob et al., 2001), which was run with an hourly time-step driven by the six hourly boundary conditions obtained from the ERA-Interim re-analysis (Dee et al., 2011). The resolution of REMO in our set-up is 0.1667 degrees. The JSBACH regional run had the same spatial and temporal resolutions. In this study, we focused on the region 52° – 72°N, 4° – 32°E.

We made comparisons between modelled and observed SIF and also between modelled SIF and GPP and the MPI-BGC GPP product that is available at monthly time scales (Jung et al., 2009, 2011). The GPP product is data-based and has been upscaled for regional and global scales using the model tree ensemble approach. For the analysis the model grid points with less than 50% of vegetation cover were omitted.

### 3 Results

### 3.1 Annual time series of ChlF and GPP at leaf and site scale

Observed ChlF and GPP had a pronounced annual cycle at Hyytiälä forest (Fig. 1). The observed quantum yield of photosynthesis $\Phi_p$ decreased to a low winter level later than the observed fluorescence signal F'. The F' started to decrease around doy 280, later than the observed GPP flux, which was reduced to zero around mid-November The increase in observed ChlF variables; F' and $\Phi_p$, took place at the same time in spring and this was also connected to the beginning of photosynthesis.

Simulated SIF decreased earlier in autumn than the other ChlF variables. This was likely associated with the decline in incoming radiation. The simulated GPP was a good match with observations during autumn. The simulated $\Phi_{f,s}$ declined simultaneously with the observed ChlF variables, however, from November to beginning of February it was on a lower level compared to them. This might be connected to the way electron transport rate (ETR) was simulated in the model. In the Farquhar model the temperature dependency of the parameter has a lot of influence on ETR, whereas the observed ChlF variables gave reason to suggest that the ETR stayed on a higher level later to the winter. The gradual decline of the observed ChlF variables might be due to dark autumns, as the needles do not suffer from excess light levels and thus the downregulation of the light harvesting machinery can be much lower (Kolari et al., 2014). The yield of photochemistry and fluorescence declines during winter because the yield of NPQ increased in a process regulated by air temperature (Porcar-Castell, 2011).

The observed ChlF variables drop to their lowest value in February and March. This was the time period, when the forest was experiencing stress because of the increasing light levels, but still persisting soil freeze and low air temperatures. The trees will have need for photoprotection in order to get rid of the excess light energy that they cannot yet use for photosynthesis, because of the prevailing conditions. Therefore the observed ChlF values obtained their lowest values in this time period. The low values in the simulated ChlF variables are caused only by low temperatures, the mechanisms related to photoprotection are not included in the current model implementation.

Simulated $\Phi_{f,s}$ showed an earlier ascent in spring, and this was connected to simulated photosynthesis that commenced too early, clearly seen in the half-hourly GPP values in Fig. 1b. The comparison between simulated $\Phi_{f,s}$, SIF and observed incoming PAR showed that in spring $\Phi_{f,s}$ that slowed down the increase of SIF to its summertime values, whereas in autumn light limitation caused the withdrawal of SIF (Supplement A, Fig. A1).

Some negative GPP values are present in Fig. 1. The random nature of turbulence and instrument uncertainty add to measurement uncertainty (Rannik et al., 2016). The GPP is obtained from the observed net ecosystem exchange (NEE) by subtracting the respiration that has been estimated by a regression fit to temperature (Wohlfahrt and Galvano, 2017). Thus the random measurement error leads to some negative GPP values that are compensated by equal amount of too high positive values, additionally the temperature fit to respiration causes some systematic error in the values.

Observed F' and $\Phi_p$ decreased at high light levels on a sunny day (Supplement A, Fig. A2). While this decline also occured in simulated $\Phi_{f,s}$, simulated SIF is increased with incoming radiation. On a cloudy day, observed F' increased during the day, whereas $\Phi_p$ showed some decrease from the morning value (Supplement A, Fig. A3). Simulated $\Phi_{f,s}$ and SIF both increased during daytime under favorable photosynthesis

conditions. This is in agreement with the expected inverse relationship between $\Phi_f$ and $\Phi_p$ under low light conditions and the positively correlated relationship under high light conditions.

## 3.2 Upscaling to site scale

In JSBACH, most of the ChlF signal originated from the top of the canopy, as it received most of the light and, therefore, the largest part of photosynthesis takes place here (Fig. 2). On a sunny day around midday, 86% of total canopy GPP and 88% of SIF was produced in the uppermost layer. On a cloudy day 97% of the total canopy GPP and 98% of the total canopy SIF was generated in the uppermost layer.

## 3.3 Comparison of remote sensing results at site scale

Overall, the remotely sensed SIF signal followed the seasonal cycle of observed GPP and modelled SIF and GPP at the flux sites (Fig. 3). Observed and modelled SIF showed a larger correlation to observed and modelled GPP, respectively, than fAPAR, in particular the FI-Hyy and FI-Ken sites (Table 2). The model was better at predicting the observed GPP than observed SIF (Table 2), which might reflect the scale mismatch of the SIF observations. Nevertheless, the ability of JSBACH to simulate fAPAR was not as good as its simulation of the SIF signal (Table 2).

The modelled GPP had the tendency to increase too early in spring, which was clearly seen at FI-Kns (Fig. 3a) and FI-Sod (Fig. 3e). This increase happened shortly before the start of the observed photosynthesis. This early emergence of photosynthesis contributed to the inability to simulate the observed SIF signal.

The slope between modelled GPP and SIF was 8.7 g C m$^{-2}$ day$^{-1}$/(unitless) (standard deviation 0.5 g C m$^{-2}$ day$^{-1}$/unitless) averaged over the four sites (Table 3). It is close to the slope between observed GPP and SIF averaged over the four sites, which was 8.9 g C m$^{-2}$ day$^{-1}$/ mW m$^{-2}$ sr$^{-1}$ nm$^{-1}$ (standard deviation 1.0 g C m$^{-2}$ day$^{-1}$/ mW m$^{-2}$ sr$^{-1}$ nm$^{-1}$). Despite this similarity of averaged values, the slopes between the model and simulations were not within the uncertainty at any other site than FI-Hyy and their results are not directly comparable, as the units are different. However, the standard deviations were lower for GPP vs. SIF fits than for fAPAR fits, when compared to the absolute values of the slopes. The slope between GPP and fAPAR in the observations was higher for the southern sites compared to the northern sites and the same was observed for the modelled slopes. The differing slopes between simulations and observations for the GPP vs. fAPAR fits resulted from differing ranges in the simulated and observed fAPAR values. The southern sites had higher GPP vs. fAPAR slopes, since the GPP values at the southern sites had much higher summertime values than the northern sites, although the fAPAR values showed a similar range at all the sites.

### 3.3.1 Year 2009 at FI-Ken

Modelled GPP and modelled SIF were well correlated with each other at all sites except FI-Ken (Table 2), which showed decoupling of these two variables in summer 2009 (Fig. 3g). In contrast to the observations, the model predicted a drought-related decline in summertime GPP in 2009. This might be due to the presence of a thin humus layer at FI-Ken, which probably makes the site more resistant to drought (Hillel, 1980), whereas JSBACH is only able to simulate freely draining upland soils.

The ensuing decoupling of modelled SIF and GPP variables was connected to the current formulation of the actual electron transport rate ($J_a$) in the model (eq. (A6)). The formulation of eq. (A6) states that the actual electron transport rate is $J$ from eq. (2) when photosynthesis is limited by the electron transport rate.

At FI-Ken the simulated summer drought influenced the simulated GPP via soil moisture limitation in the stomatal conductance. In the JSBACH model, soil moisture limitation causes a reduction in both the electron transport and the carboxylation rate limited branches of photosynthesis. This is different from other models, such as SCOPE, in which drought causes an additional decrease in $V_{c(max)}$ that further drives down the carboxylation rate limited carbon assimilation $A_c$ and results in a shift to carboxylation rate limited photosynthesis. Closer examination of the FI-Ken simulation results revealed that it was mostly dominated by the electron transport rate limited photosynthesis, therefore the drought effect seen in simulated GPP did not lower SIF.

### 3.4 Regional runs

The correlation between observed and simulated SIF for the averaged five year period was reasonably high ($r^2$=0.86) for different grid points in the study region. The correlation between simulated GPP values and the MPI-BGC GPP product was at a similar level ($r^2$=0.78). Overall, simulated GPP values were lower than the estimates from the data-driven MPI-BGC GPP, with the highest simulated GPP values less than 1200 g C m$^{-2}$ year$^{-1}$ while most of the grid points located south of 58°N according to the MPI-BGC GPP have larger values. The distribution of GPP on the map showed that the MPI-BGC GPP-product predicts much larger GPP values for the Norwegian coast than JSBACH (Fig. 4). During winter months, larger GPP values than in the surrounding regions were also seen in the MPI-BGC product (Fig. 4p). The low values in that region in JSBACH originated from the vegetation maps used for the generation of the PFT distribution. A pronounced difference between the model results and observations is that the simulated GPP reached the zero level, whereas this did not occur for the MPI-BGC GPP-product, which was not below 380 g C m$^{-2}$ year$^{-1}$ in our study region.

Modelled GPP and MPI-BGC GPP-product showed a similar pattern as the observed and modelled SIF along a longitudinal transect at 28°E with little difference in elevation (Fig. 5). At lower latitudes, MPI-BGC GPP was lower, which was not evident in the modelled variables. The observed SIF had maximum at higher latitude than other shown variables and shows therefore also lower level in high latitude values. At high latitudes (> 69.5°), the estimates from JSBACH dropped noticeably compared to the MPI-BGC GPP product.

During spring, satellite observed SIF showed a number of larger values in central Finland (Fig. 4a) that were not seen in the model variables (Fig. 4b-d). In summer, satellite observed SIF showed a larger gradient in the north-south direction than the model variables (Fig. 4e-h). This might reflect the fact that the observed gradient in green biomass was larger than seen in the simulations (Markkanen et al., in preparation). In autumn, the geographical distribution was quite similar between observed and modelled SIF and GPP variables (Fig. 4i-l). This might be connected to the

strong light dependence of SIF and GPP, both in real world and simulations, as light is a very important driver for photosynthesis in autumn. At winter, satellite observed SIF showed some scattering in the area where it had values (Fig. 4m). These values were likely connected to the challenges of winter time measurements (e.g. low light levels) with GOME-2.

At the seasonal scale, the strongest correlation ($r^2$=0.87) between satellite observed and simulated SIF occurred in autumn (September-November)

(Fig. 6c). The high correlation during autumn is likely related to the inherent light dependency of both GPP and SIF as light diminishes along latitudinal gradient. The slope of the fit between observed and modelled SIF values changed in spring compared to the summer and autumn periods. This was caused by our large latitudinal gradient of the region. In the southernmost region there appears to be some linear dependency between modelled and observed SIF, but in the northernmost region the modelled SIF values are still very close to zero.

The seasonal cycle at a monthly resolution for the different latitudinal regions revealed differences between the modelled and observed SIF (Fig. 7a). The highest SIF in the simulations occurred in July in all latitudinal regions. However, the highest value in the observations in low latitude regions occurred in June while in region 62-66°N highest value took place in July and north of 66°N the highest value occurred in August. Two of the studied micrometeorological measurement sites were located in the northernmost latitudinal region. At FI-Ken, observed SIF predicted the highest activity one month later than the observed GPP. As with simulated SIF, simulated GPP from JSBACH showed the highest value in July,

even though in the two most southern regions June and July were at a similar level (Fig. 7b). The highest values of GPP in Denmark occur in June due to the cultured crops (Lansø, 2016) and similar crops might also influence seasonal cycle in the Baltic region. The GPP from MPI-BGC was similar to the satellite observed SIF highest value in the southernmost region in June and for the other regions the highest values occurred in July (Fig. 7b).

## 4 Discussion

### 4.1  Site level observations

The implementation of the SIF leaf-scale model into JSBACH performed appropriately when compared to observations from FI-Hyy, a typical coniferous site for southern Finland. The fact that both simulated GPP and ChlF increased earlier than their observed counterparts in spring would suggest that ChlF observations might be successful in improving modelling of photosynthesis, e.g., in a data assimilation set-up (Koffi et al., 2014; Norton et al., 2017). However, it should also be noted, that the data for FI-Hyy was derived from active measurements, and that the coupling between $\Phi_p$ and $\Phi_f$ might be changed during different seasons (Krivosheeva et al., 1996). Active measurements have different variables than the "passive" quantities obtained from the model, and therefore, there is reason to be cautious with the comparison.

The SIF and fAPAR are close to each other in observations, as they are both related to green biomass. However, in the JSBACH model their calculation is different with fAPAR derived as a function of LAI and radiation, whereas GPP (and therefore SIF) is a function of other environmental variables and model parameters (in addition to LAI and radiation) that may also have an effect.

The photosynthesis of forests is often modelled using constant temperature response for the biochemical model parameters $V_{max}$ and $J_{max}$ throughout the year. However, studies have revealed that this assumption does not hold for ecosystems with strong seasonal cycles, but causes overestimation of $CO_2$ fluxes in transition periods, at least in spring. Kolari et al. (2014) found seasonally varying values for leaf level for those parameters from leaf level observations at FI-Hyy. Ueyama et al. (2016) found seasonally varying biochemical model values at four different black spruce forests in Alaska in a model inversion study at eddy covariance sites. In an earlier study using inversion at boreal coniferous forests (Thum et al., 2008), it was found that three forests at northern boreal zone (FI-Hyy, FI-Sod and FI-Ken) had temporal evolution in the biochemical parameters, but a site located on temperate boreal (Norunda, Sweden) did not.

Leaf level studies have used temperature acclimation for the changes of biochemical parameters (Wang et al., 1996). Similar results have been obtained for site level results at FI-Sod, where dark acclimated chlorophyll fluorescence observations have been used in combination with eddy covariance observations to disentangle the effect of changing maximum photosynthetic capacity (Thum et al., 2017).

The changes taking place in the needles of conifer forests in winter are numerous to protect the needles in challenging environmental conditions. E.g. the light harvesting complexes are aggregated (Porcar-Castell, 2011) and the xanthophyll cycle enables photoprotection (Ensminger et al.,

2004). Some of these processes can be in future be included in a large scale model, as adding changes to the parameters in the ChlF model discussed below, but as changes in the boreal spring happen at quite fast pace and those can be tracked with several different environmental and biological variables (Thum et al., 2009), for large scale applications a temperature related changing of the biochemical parameters might be next step forward and remotely sensed SIF observations provide a very useful evaluation tool in this context."

The number of active PSII reaction centers (parameter $q_{Ls}$ in the chlorophyll fluorescence model) has been shown to change seasonally in boreal environments (Porcar-Castell, 2011). However, in our implementation we assumed it to be a constant 0.5, as there is no theory to predict variations of this parameter at larger scales. Similarly the rate constant of sustained thermal dissipation (parameter $k_{NPQs}$ in the chlorophyll fluorescence model) incorporates seasonal variation in boreal forests (Porcar-Castell, 2011), but for the same reasons it was kept as zero in our

model runs. The comparison with the data nevertheless suggests that these assumptions are justified at the time and spatial scales of our analysis.

However, since the seasonal cycle was captured quite well by the model at FI-Hyy, even though the seasonally variable parameters that control yield were not considered, some concerns connected with the model are evident. The link to the Farquhar model causes the simulated ChlF variables to have a pronounced seasonal cycle similar to the measurements, even though the light reactions of the SIF model do not include the

seasonal changes that take place in the leaves. While this could be considered as a counterargument against our approach, the fact that we can generate an appropriate time series with the environmental controls of the Farquhar model suggests that our approach maybe a sensible choice when attempting to simulate SIF at ecosystem and larger scales.

## 4.2  Satellite data

The satellite SIF observations have a clear-sky bias, which may affect the seasonality of these data. Furthermore, illumination and viewing

geometries affect the observed SIF (Joiner et al., 2013). Also, the low SIF values measured at high latitudes make the data over those regions prone to systematic errors, which may affect the consistency of the time series. In addition, the illumination-observation geometry might play a role in canopy structure effects and its seasonality.

Here we compared site level observations with satellite observations, despite the fact that these two observations are at completely different

spatial scales. A flux tower measures approximately 1 km$^2$ of the surrounding area, whereas the satellite observations have a wider footprint (e.g. for GOME-2 default footprint is 80 km x 40 km). However, Finnish territory consists of large areas of forest and the seasonal cycle is driven by meteorological variations that have an influence at larger spatial scales, and therefore we consider the comparison between these different scales to be appropriate assuming a homogeneous landscape.

At large scale the ability of SIF to estimate the seasonal cycle of GPP has been shown at boreal coniferous forests (Walther et al., 2016). At the leaf scale the connection is more complex. The seasonal dynamics of interplay between ChlF and photosynthesis still remain unclear, and a model that captures that relationship is not currently available (Porcar-Castell et al., 2014). If alternative electron sinks or metabolic pathways exist, as was found by Krivosheeva et al. (1996) for wintering Scots pines, then this may mean that the use of ChlF as a proxy for seasonal dynamics of GPP is problematic (Porcar-Castell et al., 2014). Development of process-based models for ChlF is ongoing and once a suitable leaf-level model that incorporates the seasonal changes in the ChlF becomes available, then it could be used to parameterize the larger scale models.

## 4.3 Challenges in modelling: Radiative transfer

A significant challenge to the comparison of modelled and observed SIF is the radiative transfer in the canopy. However, as most of the detectable ChlF signal originates from the topmost canopy layer (van der Tol et al., 2014), a complicated radiative transfer scheme is not essential for a first order comparison that focuses on the seasonal cycle and large-scale gradients. This assumption is consistent with our model, which predicts that the largest part of the SIF signal originates from the topmost layer of the canopy. Therefore, we would suggest that our simplifications in treating the radiative transfer of SIF in the canopy are adequate for the purpose of this study.

## 5 Conclusions

SIF in a northern coniferous forest occurred simultaneously with GPP in both observations and simulations across Finland and the Fenno-Scandinavian region. Site level comparisons to flux tower observations of GPP support these results. The leaf level measurements provided the first comparison to simulations and it is also essential that site-level SIF observations are available, such as in study by Yang et al. (2015). A measurement set-up is currently being tested at FI-Sod that will also provide data suitable for modelling purposes.

The main findings of our study include:

- JSBACH was better in simulation of SIF than fAPAR at the site scale.
- Observed SIF was better at capturing the seasonal cycle at the forest sites than the modelled SIF and GPP, therefore it can be used to constrain modelled SIF in order to improve the simulated GPP.
- Correlation between observed SIF and observed GPP was higher in the southern than in the northern sites.
- Slopes of regression between GPP and SIF were similar between simulations and observations across different sites.

- Slopes of regression between observed GPP and fAPAR were higher in the southern than in the northern sites, and the same trend occurred for the simulated values.
- Satellite observed SIF showed a maximum seasonal value in July for the area north of latitude 66°, in contrast to the simulated SIF and simulated and observed GPP values.

Further evaluation of these results would benefit from the additional use of other remote sensing products for LAI estimates, as LAI has a strong influence on the spatial variation of the SIF signal. Current and future space missions (e.g. Guanter et al., 2014) as well as increased ground and airborne SIF observations will further provide data to relate SIF to photosynthesis.

**6 Code availability**

10 The SCOPE model is available from Christiaan Van Der Tol. The chlorophyll fluorescence model is available from Federico Magnani. The JSBACH model is available to the scientific community under a version of the Max Planck

Institute for Meteorology Software License Agreement (http://www.mpimet.mpg.de/en/science/models/license/).

**7 Data availability**

The leaf level chlorophyll fluorescence data is available from Albert Porcar-Castell and will be available from

15 http://avaa.tdata.fi/web/smart/smear during winter 2016-2017. The micrometeorological data and meteorological data is available from Annalea Lohila (FI-Kns), Mika Aurela (FI-Ken and FI-Sod) and at http://avaa.tdata.fi/openida/dl.jsp?pid=urn:nbn:fi:csc-ida-2x201611242015017385197s for FI-Hyy.

**8 Appendices**

**Appendix A: Leaf-level chlorophyll fluorescence model**

20 The leaf level model for ChlF is based on work by Magnani and Dayyoub (2016). The definitions of the variables and parameters and their possible numerical values and references are in Table A1.

Excitation energy that enters the leaf will be dissipated through photochemistry (subscript *p*), fluorescence (*f*), energy-independent (*D*) and energy-dependent heat dissipation (*NPQ*) with the following yields (*Φ*) calculated with rate constants $k_i$:

$$\Phi_p = \frac{k_p}{k_p+k_f+k_D+k_{NPQ}} \tag{A1a}$$

$$\Phi_f = \frac{k_f}{k_p+k_f+k_D+k_{NPQ}} \tag{A1b}$$

$$\Phi_D = \frac{k_D}{k_p+k_f+k_D+k_{NPQ}} \tag{A1c}$$

  $$\Phi_{NPQ} = \frac{k_{NPQ}}{k_p+k_f+k_D+k_{NPQ}} \tag{A1d}$$

The rate constant of photochemistry ($k_p$) can be expressed as a function of the intrinsic rate of photochemistry ($k_{PSII}$), photochemical quenching parameter $qL_T$ (representing the fraction of functional and open reaction centers) consisting of $qL_r$ (the fraction of open reaction centers) and $qL_s$ (the fraction of functional reaction centers, the sustained component of the photochemical quenching parameter):

$$k_p = k_{PSII} \cdot qL_T = k_{PSII} \cdot qL_s \cdot qL_r \tag{A2}$$

The rate constant for regulated thermal energy dissipation ($k_{NPQ}$) consists of reversible component (*NPQs*) and sustained component (*NPQr*):

  $$k_{NPQ} = k_{NPQs} + k_{NPQr} \tag{A3}$$

The fluorescence and photochemistry yields can be expressed by combining equations (A1-4):

$$\Phi_p = \frac{k_{PSII} \cdot qL_s \cdot qL_r}{k_{PSII} \cdot qL_s \cdot qL_r + k_f + k_D + k_{NPQr} + k_{NPQs}} \tag{A4a}$$

$$\Phi_f = \frac{k_f}{k_{PSII} \cdot qL_s \cdot qL_r + k_f + k_D + k_{NPQr} + k_{NPQs}} \tag{A4b}$$

and the ratio of these two gives:

$$\frac{\Phi_p}{\Phi_f} = \frac{k_{PSII}}{k_f} \cdot qL_s \cdot qL_r \qquad (A5)$$

The actual electron transport $J_a$ is

$$J_a = J(I) \cdot \frac{A}{A_j} \qquad (A6)$$

10 where $A$ is photosynthesis (minimum of the electron transport rate limited photosynthesis $A_j$, and maximum carboxylation rate limited photosynthesis, $A_c$, in units µmol m$^{-2}$ s$^{-1}$), $J$ is the electron transport shown in eq. (2). The $J_a$ is used in the calculation to describe the fraction of incoming quanta that is used for photosynthesis, i.e. the photochemical quantum yield of photosystem II (PSII), $\Phi_p$

$$\Phi_p = \frac{J_a}{I} \qquad (A7)$$

The rate of PSII reduction can be assumed to be proportional to the fraction of functional and closed reaction centers:

$$J_a = qL_s \cdot (1 - qL_r) \cdot J_{max} \qquad (A8)$$

20 Therefore, the fraction of reaction centers that are functional and open is:

$$qL_s \cdot qL_r = qL_s - \frac{I \cdot \Phi_p}{J_{max}} \qquad (A9)$$

By substituting eq. (A9) to eq. (A5), the following expression for fluorescence yield is obtained:

25 $$\Phi_{f,1} = \Phi_p \cdot \frac{k_f}{k_{PSII}} \cdot \frac{1}{qL_s - \frac{I \cdot \Phi_p}{J_{max}}} \qquad (A10)$$

The rate constant $k_{NPQ}$ is constant or close to zero in conditions of light-limited carboxylation (Walters et al., 1993), as energy dependent heat dissipation is the result of pH build-up in the thylakoid lumen and xanthophyll de-epoxidation. From eq. (A1b) and (A1d) the thermal energy dissipation in low light conditions is:

5 $$\Phi_{NPQ} = \Phi_f \cdot \frac{k_{NPQs}}{k_f} \qquad (A11)$$

From eq. (A1b) and (A1c) we obtain:

$$\Phi_D = \Phi_f \frac{k_D}{k_f} \qquad (A12)$$

Under these conditions a negative relationship between photochemical and fluorescence yields is expected, since:

$$\Phi_p = 1 - \Phi_f - \Phi_D - \Phi_{NPQ} = 1 - \Phi_f - \Phi_f \cdot \frac{k_D}{k_f} - \Phi_f \frac{k_{NPQs}}{k_f} \qquad (A13)$$

15 From this equation fluorescence yield at low light conditions can be derived to be:

$$\Phi_{f,2} = (1 - \Phi_p) \cdot \left( \frac{k_f}{k_f + k_D + k_{NPQs}} \right) \qquad (A14)$$

The fluorescence yield of PSII, $\Phi_f$, is taken as the minimum of $\Phi_{f,1}$ and $\Phi_{f,2}$:

$$\Phi_f = \min \left( \Phi_{f,1}, \Phi_{f,2} \right) \qquad (A15)$$

The reference minimum fluorescence yield, obtained in dark-acclimated foliage in the absence of stress $\Phi_{f,0}$ can be theoretically derived from the rate constant of fluorescence $k_f$ ($6.7 \cdot 10^7$ s$^{-1}$) (Rabinowich and Govindjee, 1969), the rate constant of thermal deactivation $k_D$ ($6.03 \cdot 10^8$ s$^{-1}$) and 25 the rate constant for photochemistry in open PSII reaction centers $k_{PSII}$ as

$$\Phi_{f,0} = \frac{k_f}{k_f + k_{PSII} + k_D} \tag{A16}$$

and $k_{PSII}$ (Genty et al., 1989) can be derived as

$$k_{PSII} = \frac{(k_D + k_f) \cdot \Phi_{p,max}}{(1 - \Phi_{p,max})} \tag{A17}$$

where $\Phi_{p,max}$ (0.88 mol / E) is the maximum quantum yield of PSII in dark-acclimated conditions in the absence of stress obtained fluorometrically after correction for PSI fluorescence (Pfundel, 1998).

To obtain the scaled fluorescence yield $\Phi_{f,s}$, the fluorescence yield $\Phi_f$ is further divided by $\Phi_{f,0}$,

$$\Phi_{f,s} = \frac{\Phi_f}{\Phi_{f,0}} \tag{A18}$$

## 9 Author contribution

T. Thum and S. Zaehle did the implementation of the chlorophyll fluorescence model into the JSBACH model. P. Köhler and L. Guanter provided remote sensing data. M. Aurela and T. Laurila provided the micrometeorological and meteorological data for sites FI-Ken and FI-Sod. A. Lohila provided data for FI-Kns and P. Kolari for FI-Hyy. F. Magnani contributed the chlorophyll fluorescence model and C. Van Der Tol the SCOPE model. T. Markkanen set-up the regional model and prepared the meteorological forcing data for the model. T. Thum did the simulations and analyzed the data with help of the co-authors. T. Thum prepared the manuscript with contributions from all co-authors.

**10 Competing interests**

Author S. Zaehle is a member of the editorial board of the journal.

# 11 Acknowledgements

TT acknowledges funding from the Academy of Finland (grant number 266803). SZ was supported by the European Research Council (ERC) under the European Union's Horizon 2020 research and innovation programme (QUINCY; grant no. 647204). We thank Dr. Thomas Kaminski for providing fAPAR data for all the regions and sites. Help from Dr. Leif Backman in acquisition of ERA-Interim data is acknowledged. We thank the Academy of Finland Centre of Excellence (grant no. 1118615 and 272041). We thank Dr. Martin Jung for the use of the MPI-BGC GPP product. We are grateful to Dr. Albert Porcar-Castell for sharing his leaf-level ChlF data from Hyytiälä and for fruitful discussions. We thank Sophia Walther and Kristin Böttcher for their valuable comments on the manuscript. Two anonymous referees whose comments greatly improved the manuscript are gratefully acknowledged.

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

**Tables**

**Table 1**. Measurement sites

| | Abbreviation | Name | Location | Vegetation | total LAI [$m^2$ $m^{-2}$] | Year | Reference |
|---|---|---|---|---|---|---|---|
| 15 | FI-Kns | Kalevansuo | 60°39'N 24°21'E | Scots pine forest | 5 | 2007-2008 | Lohila *et al.* (2011) |
| | FI-Hyy | Hyytiälä | 61°51'N 24°18'E | Scots pine forest | 6 | 2007-2011 | Rannik *et al.* (2004) Mammarella *et al.* (2009) |
| 20 | FI-Ken | Kenttärova | 67°59'N 24°15'E | Norway spruce forest | 6.6 | 2007-2011 | Thum *et al.* (2008) Aurela *et al.* (2016) |
| | FI-Sod | Sodankylä | 67°21'N 26°38'E | Scots pine forest | 3.6 | 2007-2008 | Thum *et al.* (2007) |

**Table 2**. The correlation coefficient ($r^2$) and its significance (in parenthesis) between modelled and observed sun-induced fluorescence (SIF) (observed SIF in units mW m$^{-2}$ sr$^{-1}$ nm$^{-1}$) and gross primary production (GPP) (unit: g C m$^{-2}$ day$^{-1}$) values at different sites. In the calculation of linear regressions between simulated and observed GPP and SIF, 20-day time periods and the morning values were used. In the calculation of GPP vs. fraction of absorbed photosynthetic active radiation by the vegetation (fAPAR) fits, monthly values including the whole day were used.

|  | FI-Kns | FI-Hyy | FI-Sod | FI-Ken |
|---|---|---|---|---|
| Obs. GPP vs. obs. SIF | 0.91 | 0.93 | 0.89 | 0.82 |
|  | $(3.66 \cdot 10^{-19})$ | $(4.24 \cdot 10^{-51})$ | $(3.44 \cdot 10^{-11})$ | $(7.33 \cdot 10^{-22})$ |
| Mod. GPP vs. mod. SIF | 0.99 | 0.92 | 0.99 | 0.83 |
|  | $(1.20 \cdot 10^{-34})$ | $(5.80 \cdot 10^{-85})$ | $(1.04 \cdot 10^{-29})$ | $(1.38 \cdot 10^{-29})$ |
| Obs. GPP vs. mod. GPP | 0.97 | 0.98 | 0.84 | 0.82 |
|  | $(1.25 \cdot 10^{-27})$ | $(8.02 \cdot 10^{-73})$ | $(2.06 \cdot 10^{-12})$ | $(2.40 \cdot 10^{-29})$ |
| Obs. SIF vs. mod. SIF | 0.90 | 0.90 | 0.66 | 0.81 |
|  | $(4.00 \cdot 10^{-18})$ | $(3.03 \cdot 10^{-44})$ | $(3.41 \cdot 10^{-18})$ | $(9.17 \cdot 10^{-28})$ |
| Obs. GPP vs. obs. fAPAR | 0.90 | 0.66 | 0.82 | 0.72 |
|  | $(5.64 \cdot 10^{-10})$ | $(1.68 \cdot 10^{-12})$ | $(6.16 \cdot 10^{-8})$ | $(1.01 \cdot 10^{-13})$ |
| Mod. GPP vs. mod. fAPAR | 0.89 | 0.86 | 0.80 | 0.67 |
|  | $(3.01 \cdot 10^{-12})$ | $(4.98 \cdot 10^{-27})$ | $(2.33 \cdot 10^{-9})$ | $(5.82 \cdot 10^{-16})$ |
| Obs. fAPAR vs. mod. fAPAR | 0.77 | 0.70 | 0.72 | 0.67 |
|  | $(1.58 \cdot 10^{-8})$ | $(8.26 \cdot 10^{-17})$ | $(1.53 \cdot 10^{-7})$ | $(1.31 \cdot 10^{-15})$ |

**Table 3**. The slopes of the fits between gross primary production (GPP) and sun-induced chlorophyll fluorescence (SIF)/ fraction of absorbed photosynthetic active radiation by vegetation (fAPAR). Note that a constant scalar was used in multiplying the modelled SIF values.

| | FI-Kns | FI-Hyy | FI-Sod | FI-Ken | Average | Standard deviation |
|---|---|---|---|---|---|---|
| Obs. GPP vs. obs. SIF | | | | | | |
| | 7.6 (±0.4) | 9.8 (±0.3) | 9.5 (±0.7) | 7.8 (±0.5) | 8.7 | 1.0 |
| Mod. GPP vs. mod. SIF | 8.3 (±0.1) | 9.7 (±0.1) | 8.5 (±0.1) | 9.2 (±0.5) | 8.9 | 0.5 |
| Obs. GPP vs. obs. fAPAR | 22.9 (±2.5) | 24.1 (±2.5) | 13.2 (±1.5) | 12.5 (±1.2) | 18.2 | 5.3 |
| Mod. GPP vs. mod. fAPAR | 15.8 (±1.2) | 18.1 (±0.9) | 5.8 (±0.6) | 4.3 (±0.4) | 11.0 | 6.0 |

**Table A1**. Descriptions of the variables and parameters.

| Variable/Parameter (unit) | Description | Value | Reference |
|---|---|---|---|
| $\Phi_p$ | Yield for photochemistry | | |
| $\Phi_f$ | Yield for fluorescence | | |
| $\Phi_D$ | Yield for energy-independent heat dissipation | | |
| $\Phi_{NPQ}$ | Yield for energy-dependent heat dissipation | | |
| $k_p$ (s$^{-1}$) | Rate constant of photochemistry | | |
| $k_f$ (s$^{-1}$) | Rate constant of fluorescence | $6.7 \cdot 10^7$ | Rabinowich and Govindjee (1969) |
| $k_D$ (s$^{-1}$) | Rate constant of energy-independent heat dissipation | $6.03 \cdot 10^8$ | Porcar-Castell et al. (2006) |
| $k_{NPQ}$ (s$^{-1}$) | Rate constant of energy-dependent heat dissipation | | |
| $k_{PSII}$ (s$^{-1}$) | Intrinsic rate of photochemistry | | |
| $qL_T$ | Photochemical quenching parameter (representing the fraction of functional and open reaction centers) | | |
| $qL_r$ | The fraction of open reaction centers | | |

| | | | |
|---|---|---|---|
| $qL_s$ | The fraction of functional reaction centers | | |
| $k_{NPQr}$ (s$^{-1}$) | Rate constant of reversible component of $k_{NPQ}$ | | |
| $k_{NPQs}$ (s$^{-1}$) | Rate constant of sustained component of $k_{NPQ}$ | | |
| $J_a$ (μmol m$^{-2}$ s$^{-1}$) | Actual electron transport rate | | |
5 | $A$ (μmol m$^{-2}$ s$^{-1}$) | Photosynthesis | | |
| $A_j$ (μmol m$^{-2}$ s$^{-1}$) | Electron transport rate limited photosynthesis | | |
| $A_c$ (μmol m$^{-2}$ s$^{-1}$) | Maximum carboxylation rate limited photosynthesis | | |
| $J_{max}$ (μmol m$^{-2}$ s$^{-1}$) | Maximum potential electron transport rate | | |
| $\Phi_{f,0}$ | Dark-adapted fluorescence yield of PSII | | |
10 | $\Phi_{p,max}$ (mol / E) | Maximum quantum yield of PSII in dark-acclimated conditions in the absence of stress | 0.88 | Pfundel (1998) |

**Figure captions**

**Fig. 1**. a) The annual cycles (15.8.2008-14.8.2009) of measured chlorophyll fluorescence (F' and photochemical yield $\Phi_p$) and the simulated ChlF variables (SIF and fluorescence yield $\Phi_{f,s}$) from the JSBACH model at a daily scale at site FI-Hyy. All the ChlF variables are scaled from zero to one. The thick line for the modelled ChlF values is a 15-day running average of the value. b) The modelled and observed gross primary production (GPP) at half-hourly means (dots) and corresponding 30-day running mean (thick lines).

**Fig. 2**. The daily cycle of simulated SIF and $\Phi_{f,s}$ in three different canopy layers of JSBACH on day 160 in 2008. Layer 1 was the upmost layer without attenuation taken into account in a) and b) and with attenuation estimated from the SCOPE model included in e) and f). The sums of different layers are shown in c) (SIF) and d) ($\Phi_{f,s}$) with and without attenuation.

**Fig. 3**. Observed and simulated SIF and GPP scaled to unity (unitless) with corresponding standard deviations at a) FI-Kns, c) FI-Hyd, e) FI-Sod, g) FI-Ken.

**Fig. 4.** Maps for our study region with GOME-SIF, JSBACH-SIF, JSBACH-GPP and MPI BGC-GPP averaged for time period 2007–2011 and separated between seasons.

**Fig. 5.** Latitudinal transect at 28°E showing JSBACH-GPP, MPI-BGC-GPP and observed and simulated SIF averaged for the time period 2007–2008. All quantities have been scaled to unity.

**Fig. 6**. Correlation plots for different seasons, GOME-SIF vs. JSBACH-SIF. Note that JSBACH-SIF has been multiplied by 100.

**Fig. 7.** Averaged seasonal cycles of observed and simulated SIF (a) and MPI-BGC and simulated GPP (b) separated by latitudinal regions. Note that the simulated SIF value was multiplied by 100. The observed SIF is in units mW $m^{-2}$ $sr^{-1}$ $nm^{-1}$.

**Figures**

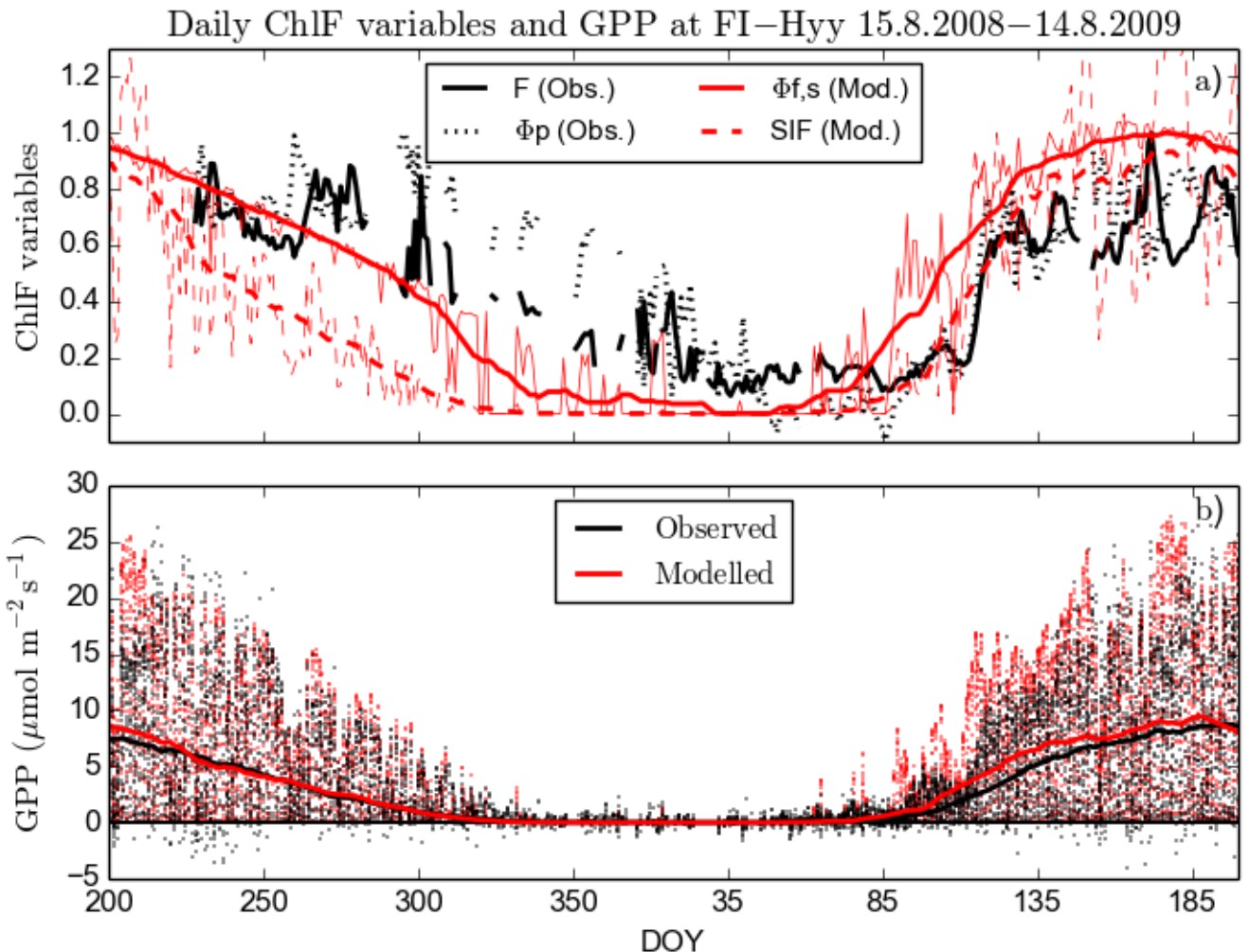

Fig. 1

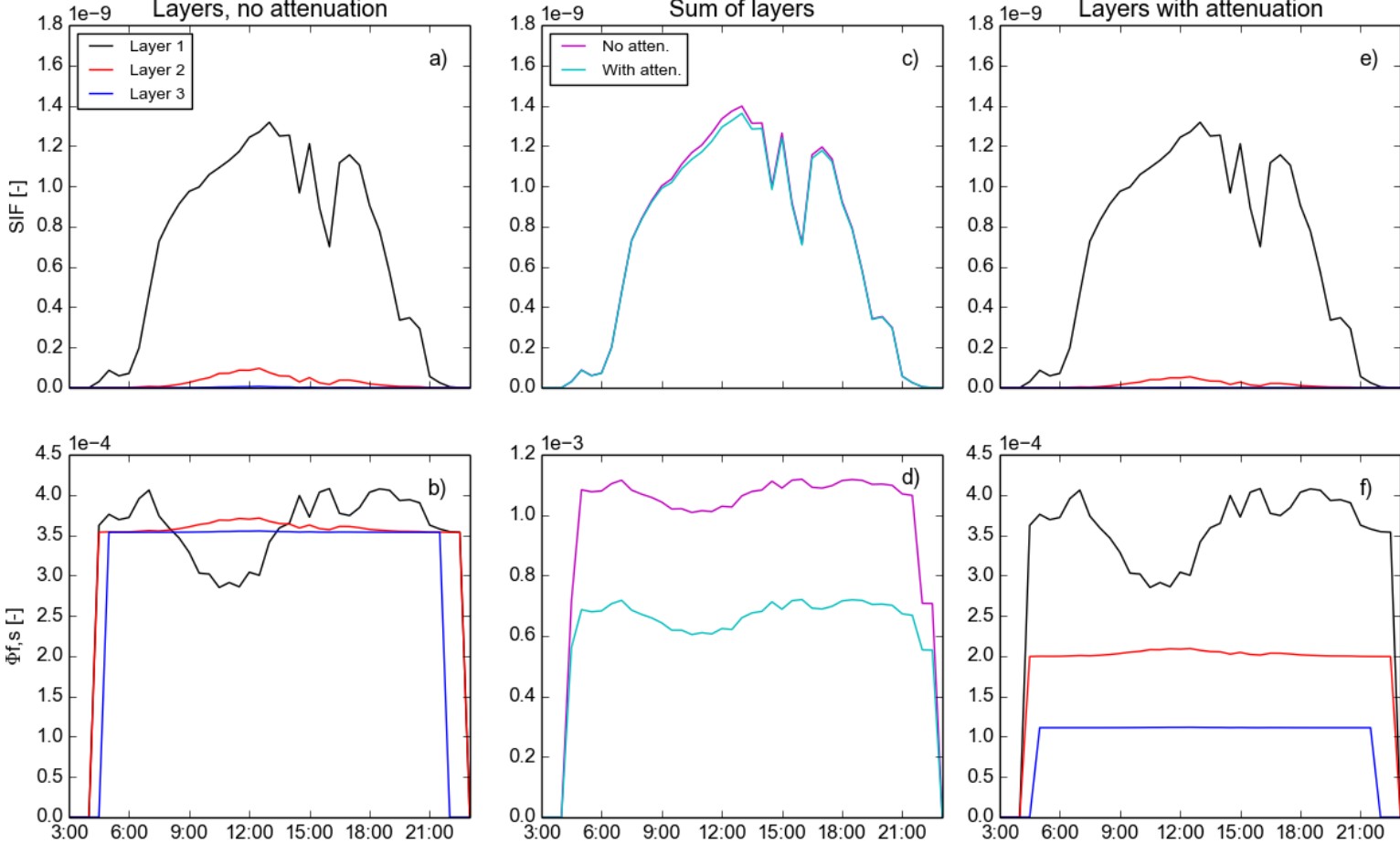

Fig. 2

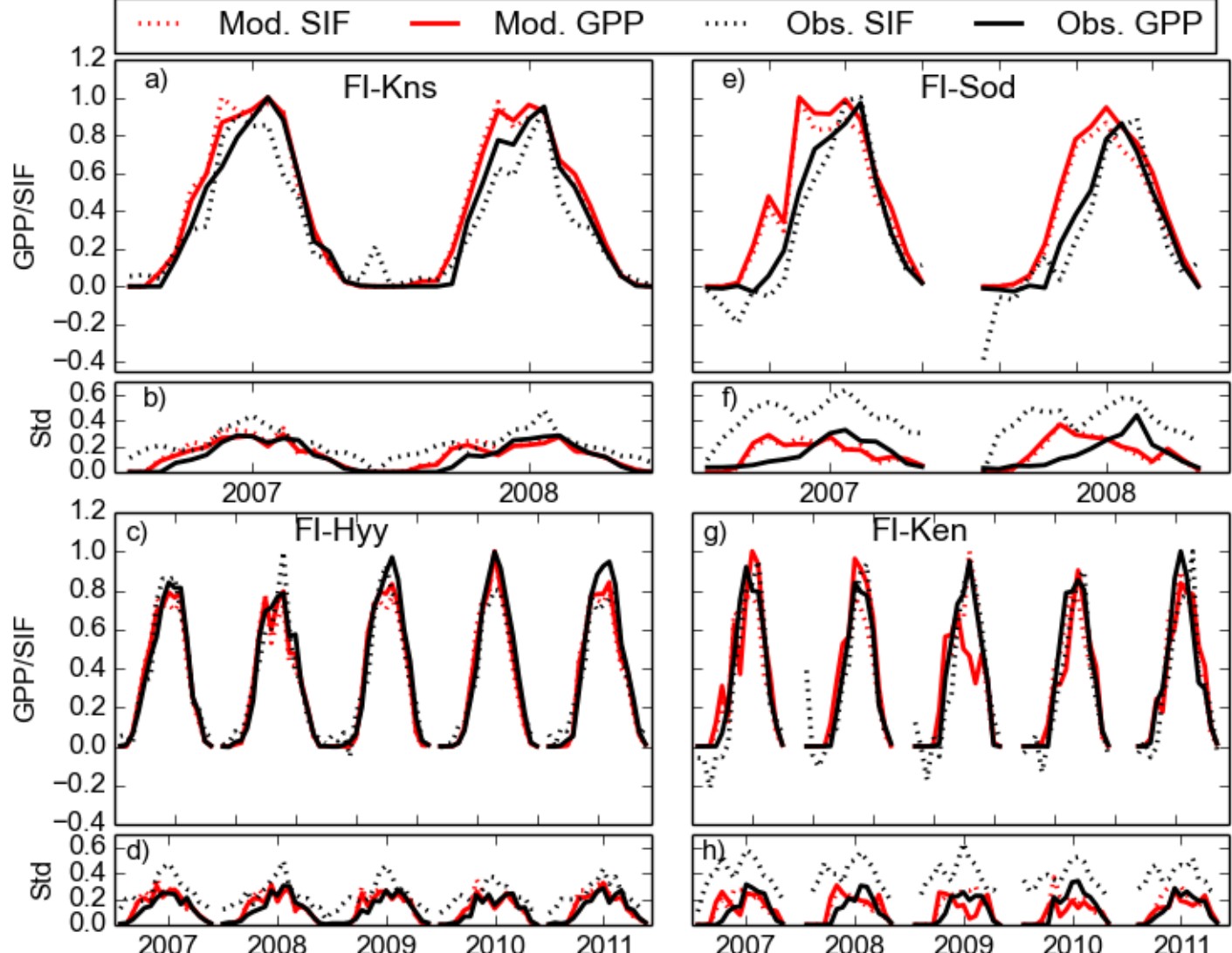

Fig. 3

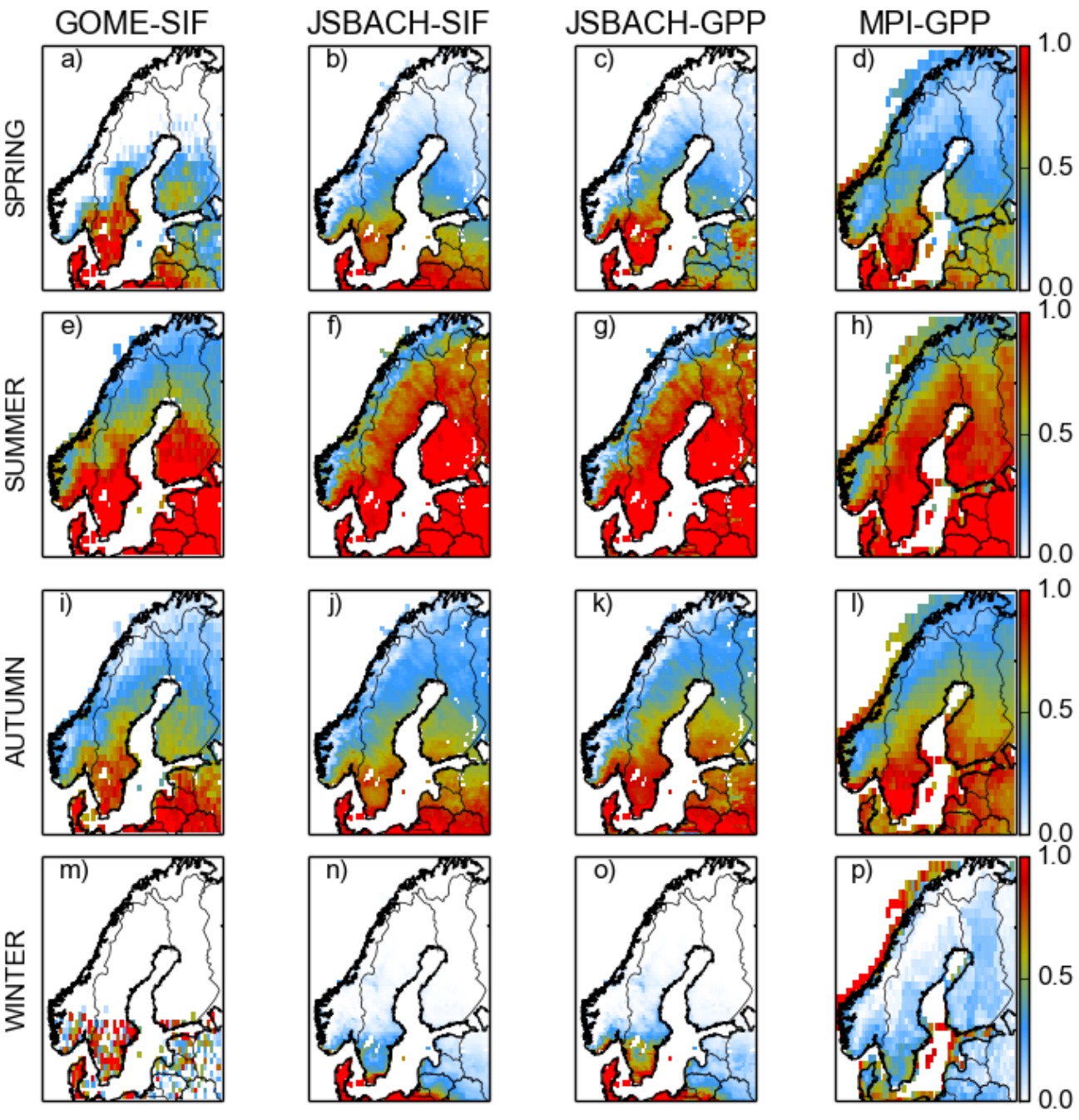

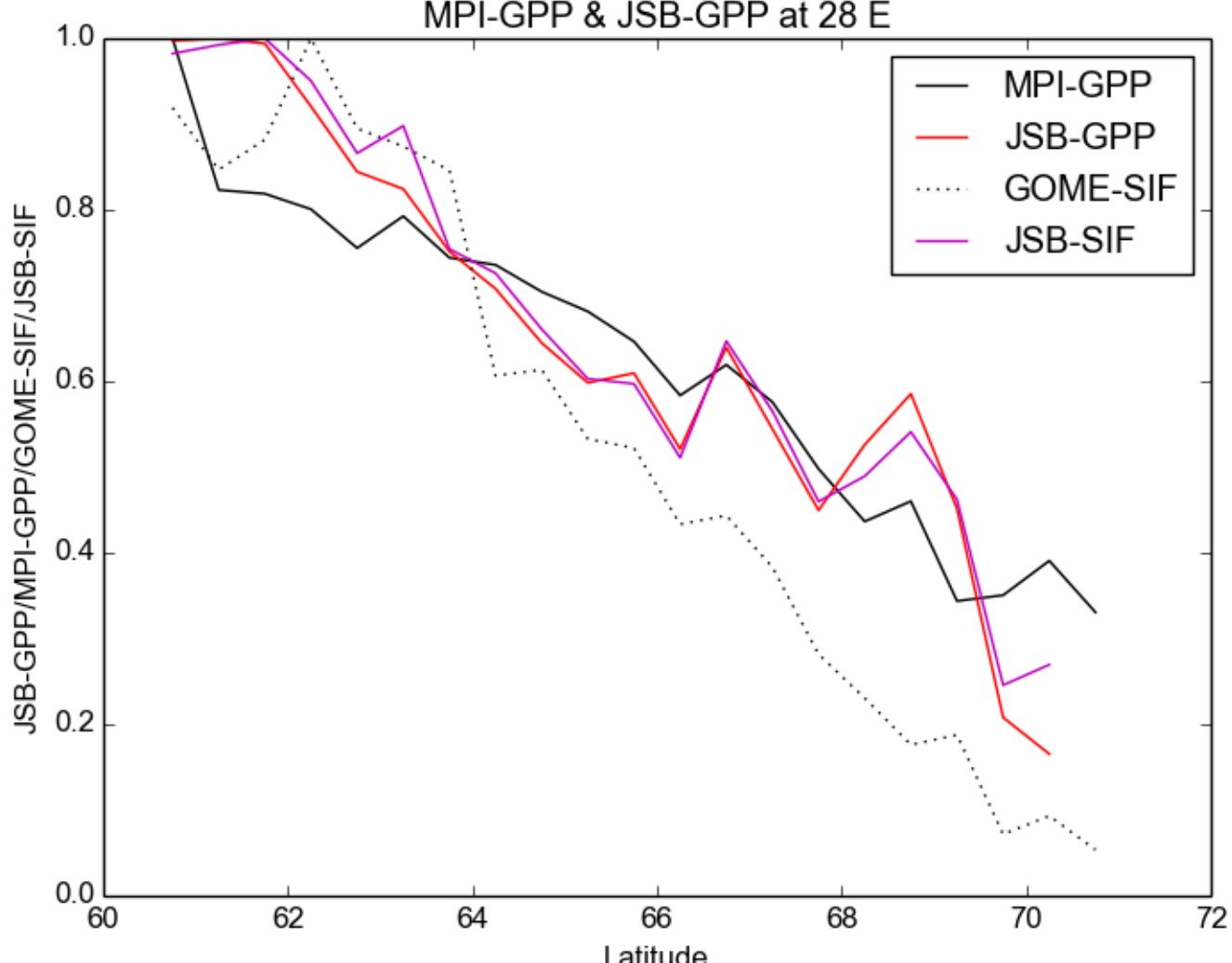

MPI-GPP & JSB-GPP at 28 E

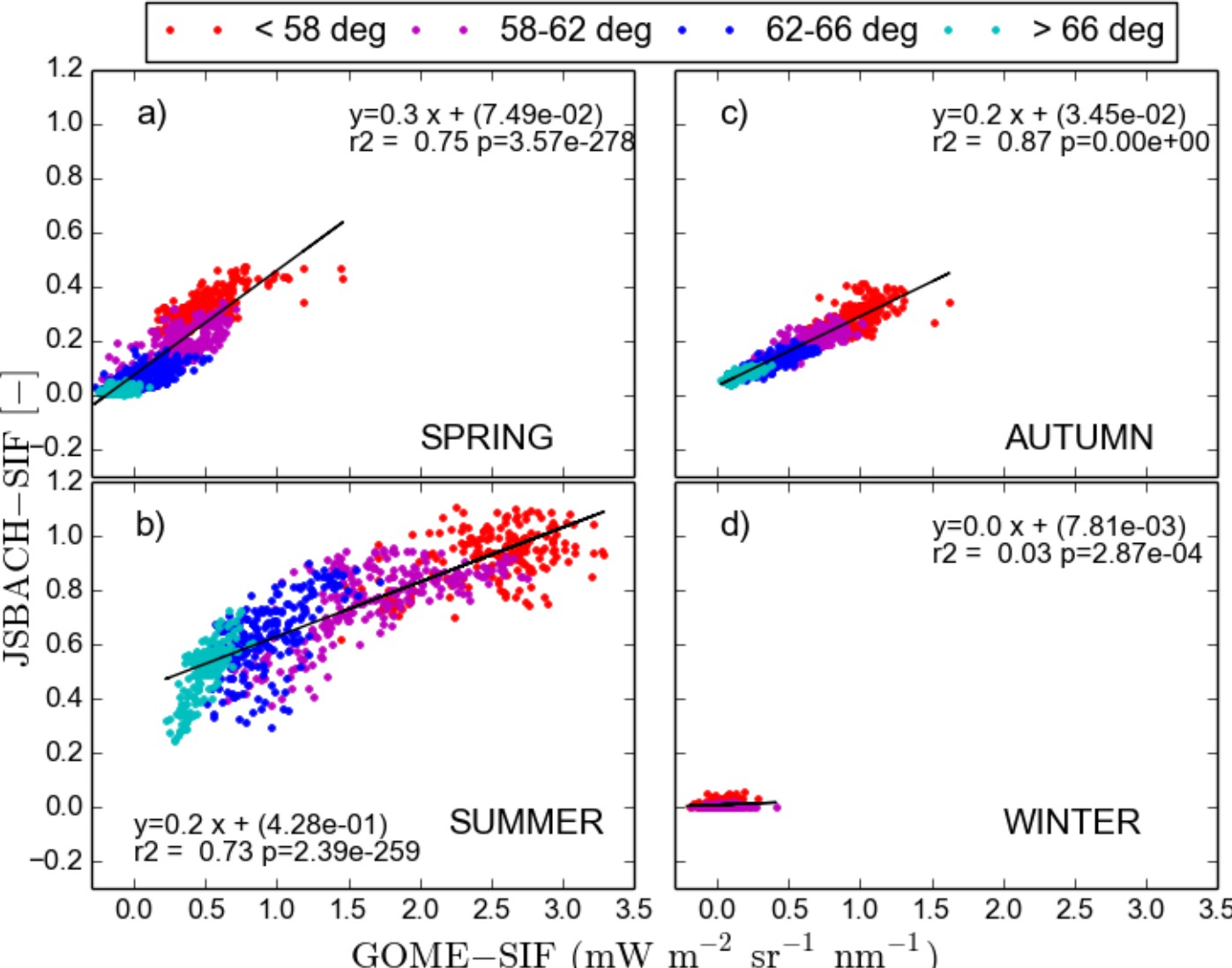

Fig. 6

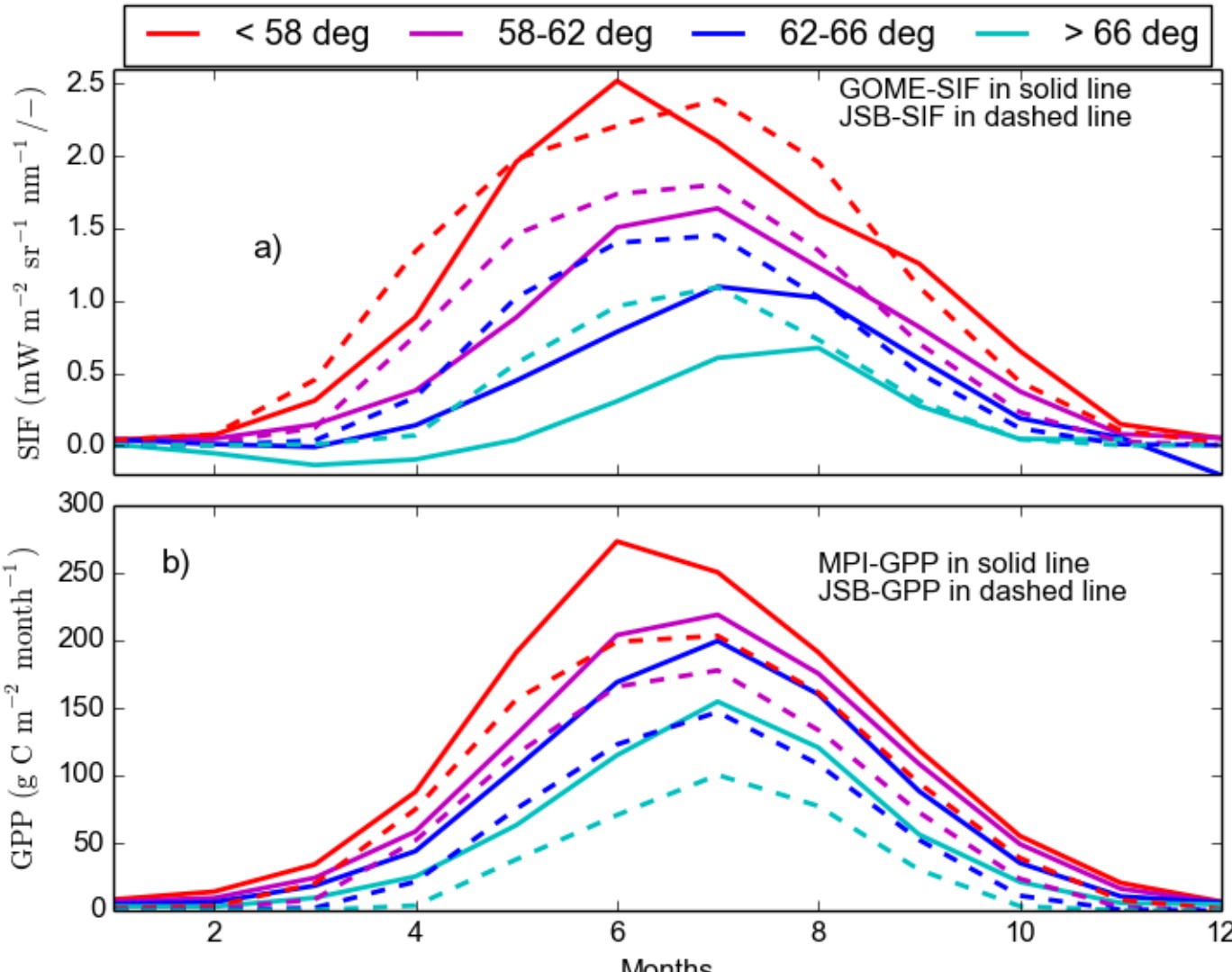

Fig. 7