# Peer review of "Modelling sun-induced fluorescence and photosynthesis with a land surface model at local and regional scales in northern Europe"

_Biogeosciences, 2016_

## Referee Comment (RC1) · Anonymous Referee #1 · 3 Jan 2017

The manuscript by Thum et al examines the use of SIF to predict GPP in coniferous forests in southern and northern Finland. The authors implement a SIF module in the JSBACH biosphere model and evaluate seasonal and spatially variability against SIF and GPP measurements at leaf, canopy, and ecosystem scale, with focus on spring and autumn transition seasons. A key innovation is the use of active leaf level fluorescence data to understand the seasonal relationship of photochemistry and fluorescence and evaluate model performance. Although many uncertainties exist in the model simulations and in understanding dependencies on environmental vs biochemical effects, the authors show good correlation of observed and simulated variables, providing some confidence for future testing and evaluation, and paving the path for

future efforts to scale between leaf and canopy/ecosystem levels. In general, I found this paper interesting and innovative, but it was hard to read at times, and the objective weren't clearly established making results and discussion hard to follow. I recommend a more careful analysis of satellite observations and some general clarifying throughout, but I expect this to be an important study with a few substantial revisions.

Major Comments

My main concern is biasing of GOME-2 time series by filtering of negative SIF values. These data are part of the noise needed in averaging to produce a smoothly varying signal. Because the noise is fixed ($\sim$0.5 mW m-2 sr-1 nm-1 as mentioned on P9 L4) and doesn't scale with signal, this technique will remove more points in fall-winter-spring when errors are large compared to signal, leading to positive cold season biases, early spring GPP onset, and underestimated seasonal amplitude. This should explain why observed SIF doesn't reach zero level (P13 L5) and why the authors find an opposite phase an opposite phase relationship of GPP and SIF in spring at FI-Hyy in active data (photochemical yield synchronized with SIF) compared to passive data where SIF precedes GPP. I recommend reanalyzing GOME-2 results with negative values.

The authors show that model GPP is systematically early in coniferous forests compared to ground and satellite data, a finding that is consistent with previous studies of cold limited ecosystems. I was hoping the authors could take better advantage of the multiscale observations and new model capabilities to provide explanations at biochemical and environmental levels, especially since the challenge of understanding the spring transition is listed as a motivation for the study. Some speculation is provided in the discussion (e.g., frost) but not much detail and no mention in the conclusions. I think this is an important enough result and application of new methods as to warrant further discussion. I would like to see the authors discuss what is needed to improve model representation of the spring transition. What would be the effect of seasonal PSII and thermal dissipation? Growing degree days, cold temperature days, and/or frozen soils?

What important environmental controls are included/missing in the Farquhar model?

I am also interested in further elaboration of results in autumn at FI-Hyy, in particular, why F' and photochemical yield are strongly delayed relative to GPP in autumn but synchronized in spring.

Minor Comments

Abstract and Conclusions – Mostly a discussion of methods and no mention of new results. I suggest discussing at least one new and interesting result from your study. Something about spring or autumn photosynthesis, or using leaf level measurements with satellite data, or comparing model simulations to active and passive fluorescence data.

Figure 1 - Figure legend is difficult to read and it's not clear from figure or caption what is being plotted in panel (A) – legend appears to suggest fluorescence yield as solid red but text refers to photochemical yield. GPP is not shown in panel (A) as stated in caption – please correct.

Figure 3 – color scheme is confusing especially with multiple variables on 1 plot. keep observations in black and models in color like in figure 1. Use same line styles for same variables (solid for GPP, dashed for SIF).

P5L10: *an indication of the fraction of electrons in the leaf that follow the ChlF pathway

P6L12: *is used in

P7L23 & P9L2: Confusion about overpass time. Here it is stated as 10:30 am but as 9:30 am in Section 2.3.2. Please clarify or correct. Also clarify what it means for the satellite overpass time to last for 100 minutes.

P10L18: I don't see the simultaneous decrease in observed GPP with F'. GPP is already declining on Day 200 while F' appears steady until ∼Day 280. F' decrease is also much more gradual and doesn't reach its minimum until January.Âă

[Figure]

P10L25: please elaborate what is meant here - are you suggesting that in low light conditions of spring, most of the absorbed radiation goes into photochemistry thus reducing that available to fluorescence? Âă

P12L1: quantify "reasonably similar" - within 10% of observations? 5%?ÂăRegression is slightly lower on average in model

P12L8: Fl-Sod has lower correlations than Fl-Ken.

P12L10: provide reference for peat effect on drought

P13L25: what is the magnitude and direction of the seasonal drift in GOME-2 overpass, and what is the likely influence?

P14L13: please explain what a static temperature response is, the effect on early GPP, and how this could be corrected in the model

P15L4: add condition "assuming a homogeneous landscape"

––––––––––––––––––––––––––

---

## Referee Comment (RC2) · Anonymous Referee #2 · 30 Jan 2017

The authors use GPP data derived from $CO_2$ fluxes measured at 4 boreal forest sites, together with SIF derived from the GOME satellite and leaf-level active fluorescence data to test a new version of the land surface model JS-Bach, which has been updated with a description of ChlF fluorescence. Finally, JS-Bach is applied at regional scale.

The authors demonstrate overall good correspondence between measured and simulated GPP (which was calibrated though) and satellite SIF and site-level GPP and reasonable correspondence to leaf-level active fluorescence data. SIF compares better to measured GPP compared to remotely sensed fAPAR.

I think this is a useful and original contribution. My comments are mostly meant to improve clarity, which the ms frequently lacks.

Detailed comments: p. 3, l. 6: as ecosystems exchange various forms of carbon, use carbon dioxide if you actually refer to carbon dioxide p. 3, l. 11: strictly speaking this is only true for fAPAR, while NDVI is just the normalized difference between reflectances in NIR and red, which happens to correlate with fAPAR p. 4, l. 1: I would contradict the "readily", given that we are still far from a truly process-based description of SIF; the Farquhar model though offers most of the interfaces for coupling to SIF p. 4, l. 12: "Both these regions ..." p. 4, l. 15-15: here you might explain why you focus on spring and autumn p. 4, l. 19: here you haven't mentioned yet that you did implement SIF into your LSM p. 4-5, section 2.1: while this section clarifies some of the basics, it entirely lacks details, such as which instrument was used for active measurements in the field and how the experimental protocol was, etc. – I see this comes later, so an appropriate header reflecting this is required here p. 6, l. 6-7: the acronyms/abbreviations do not make sense – maybe use subscripts like dir and dif to distinguish between direct (beam) and diffuse radiation; Wouldn't the equation be easier to understand if fAPAR was calculated as the difference between the radiation balance at the top of canopy (layer 1) minus the radiation balance below the lowermost layer (layer 3); replace "transmitted" by "used" or similar p. 6, l. 12: "... is used .." p. 6, l. 16: typically the temperature dependency of Jmax is either exponential or even follows an optimum shape p. 6, l. 17: isn't the value of alpha typically around 0.05 (mol CO2/mol photons) p. 7, l. 19: "obtained" – use past tense throughout p. 8, l. 19, 24: two times same header numbering p. 9, l. 5: is it a good idea to introduce a bias into the data? Isn't there some other way to deal with the negative values? p. 9, l. 9: does this explain how fAPAR is derived? I mean in the sense that a reader should be able to repeat the author's approach? p. 9, l. 18: what does "adjusted" exactly mean? Which metric did you use to measured the success of the "adjustment"? typically, Jmax is linked to Vcmax through the ratio of the two – was that done here too, i.e. only Vcmax adjusted and Jmax "followed" based on the relatively conservative ratio of the two? p. 10, l. 14: what exactly means "most" in this context? p. 11, l. 1: doesn't the term "midday depression" refer to the drought-related midday decrease in leaf net photosynthesis and

stomatal conductance? p. 16, l. 2: "wider footprint" – be more precise ... Fig. 1: might be worth commenting on the negative measured GPP values

---

## Author Comment (AC1) · 28 Feb 2017

The authors use GPP data derived from CO2 fluxes measured at 4 boreal forest sites, together with SIF derived from the GOME satellite and leaf-level active fluorescence data to test a new version of the land surface model JS-Bach, which has been updated with a description of ChlF fluorescence. Finally, JS-Bach is applied at regional scale.

The authors demonstrate overall good correspondence between measured and simulated GPP (which was calibrated though) and satellite SIF and site-level GPP and reasonable correspondence to leaf-level active fluorescence data. SIF compares better to measured GPP compared to remotely sensed fAPAR.

I think this is a useful and original contribution. My comments are mostly meant to improve clarity, which the ms frequently lacks.

We thank the referee for these comments and hope that our responses to the comments are able to clarify the manuscript.

Detailed comments:

p. 3, l. 6: as ecosystems exchange various forms of carbon, use carbon dioxide if you actually refer to carbon dioxide

Indeed, the aim was here to refer only to carbon dioxide, not to methane or VOCs, so we made this addition to clarify the text.

p. 3, l. 11: strictly speaking this is only true for fAPAR, while NDVI is just the normalized difference between reflectances in NIR and red, which happens to correlate with fAPAR

We totally agree with the referee and this was a language issue. We replaced "which" by "describing", hopefully now clarifying the issue.

p. 4, l. 1: I would contradict the "readily", given that we are still far from a truly process-based description of SIF; the Farquhar model though offers most of the interfaces for coupling to SIF

We agree with the referee. We took away the word "readily".

p. 4, l. 12: "Both these regions . . ."

Thank you, this is now corrected in the text.

p. 4, l. 15-15: here you might explain why you focus on spring and autumn

A good point. We added here the following text: "*Forests in the boreal zone experience strong seasonal cycle with cold winters and warm summers (Bonan, 2008). The transition periods of spring and autumn influence the carbon balances in these northern ecosystems (Bergh et al., 1998). In changing climate the conditions in spring and autumn will change (Ruosteenoja et al., 2011)and cause changes to carbon balances. It is anyhow during these times that the carbon cycle models have difficulties in performance (Schaefer et al., 2012). Therefore it is important to find ways to improve carbon cycle models in these time periods.*"

p. 4, l. 19: here you haven't mentioned yet that you did implement SIF into your LSM

Thank you, we changed the text so that we separately mention the implementation.

p. 4-5, section 2.1: while this section clarifies some of the basics, it entirely lacks details, such as which instrument was used for active measurements in the field and how the experimental protocol was, etc. – I see this comes later, so an appropriate header reflecting this is required here

A good point again, we added "in general" to the title in order to clarify the issue.

p. 6, l. 6-7: the acronyms/abbreviations do not make sense – maybe use subscripts like dir and dif to distinguish between direct (beam) and diffuse radiation;

Indeed, the used abbreviations were not that clear. We used referee's suggestion to distinguish between direct and diffuse radiation.

Wouldn't the equation be easier to understand if fAPAR was calculated as the difference between the radiation balance at the top of canopy (layer 1) minus the radiation balance below the lowermost layer (layer 3);

We understand the referee's point, but we have a three –layer canopy and here we wanted to show how to calculate fAPAR for each layer. As said in the text, the canopy fAPAR is the sum of fAPAR from different layers.

replace "transmitted" by "used" or similar

We made that.

p. 6, l. 12: ". . . is used .."

Thank you, corrected.

p. 6, l. 16: typically the temperature dependency of Jmax is either exponential or even follows an optimum shape

We agree, but in this study we decided not to make any changes to the original formulation of the JSBACH model, that would have required then some additional evaluations of the model performance.

p. 6, l. 17: isn't the value of alpha typically around 0.05 (mol CO2/mol photons)

We're talking here about apparent quantum yield, not the intrinsic quantum yield. We modified the text to say apparent quantum yield, which is the true quantum yield multiplied by the light absorption in the leaf (Walker et al., 2014). This value (0.28) is the default value of JSBACH for all the different plant functional types. The parameter optimization study by Mäkelä et al. (2016) done by JSBACH at site level for FI-Hyy and FI-Sod, showed this value to be good for the two sites.

Thank you, we corrected that.

Thank you, we corrected that.

p. 9, l. 5: is it a good idea to introduce a bias into the data? Isn't there some other way to deal with the negative values?

We have redone the analysis by including the negative values in the analysis.

p. 9, l. 9: does this explain how fAPAR is derived? I mean in the sense that a reader should be able to repeat the author's approach?

What the author did related to fAPAR was to ask Thomas Kaminski for these data, that he kindly provided and even took out data for each site. These values were obtained by partitioning the solar radiation fluxes that were based on inversion of the MODIS broadband white sky surface albedos and the reference for those data is given here. As we did not do the laborious processing it takes to obtain those data, we did not go into details here.

p. 9, l. 18: what does "adjusted" exactly mean? Which metric did you use to measured the success of the "adjustment"?

We didn't perform here any rigorous tuning with profound mathematical methods, like done by Mäkelä et al. (2016). Instead, we matched the averaged LAI value to the observations. We did not consider building a model tuning framework for this work, as we're not dealing with the absolute GPP values from the sites, but instead we use our modelled time series to assess the seasonal behavior of the model. We added sentence: "*No rigorous parameter inversion methods were used, as we did not use the absolute GPP values in our study, but focused more on the seasonal behavior.*" to the text.

typically, Jmax is linked to Vcmax through the ratio of the two – was that done here too, i.e. only Vcmax adjusted and Jmax "followed" based on the relatively conservative ratio of the two?

Yes, the original ratio between Vcmax and Jmax was 2.1, and we kept this same ratio, by changing first Vcmax and then always Jmax accordingly. We added this point to the text.

p. 10, l. 14: what exactly means "most" in this context?

Apologies, that was unclear. We did the analysis for points having larger fraction than 0.5 for the vegetation, but plotted all the points to the map in Fig. 4. We clarified this.

p. 11, l. 1: doesn't the term "midday depression" refer to the drought-related midday decrease in leaf net photosynthesis and stomatal conductance?

Yes, we took that away, as it's not in the right context here.

p. 16, l. 2: "wider footprint" – be more precise . . .

We added (e.g. for GOME-2 default footprint is 80 km x 40 km) in parenthesis to be more explicit.

Fig. 1: might be worth commenting on the negative measured GPP values

This is a good addition. The negative values are originating from measurement uncertainty. The GPP is obtained from the observed NEE by subtracting the respiration fitted by temperature regression. The temperature fit to respiration adds some systematic error to the GPP estimate. We wrote to the manuscript:

"*Some negative GPP values are present in Fig. 1. The random nature of turbulence and instrument uncertainty add to measurement uncertainty (Rannik et al., 2016). The GPP is obtained from the observed net ecosystem exchange (NEE) by subtracting the respiration that has been estimated by a regression fit to temperature (Wohlfahrt and Galvano, 2017). Thus the random measurement uncertainty leads to some negative GPP values that are compensated by equal amount of too high positive values, additionally the temperature fit to respiration causes some systematic error in the values.*"

References

Bergh, J., McMurtrie, R. E., and Linder, S.: Climatic factors controlling the productivity of Norway spruce: a model-based analysis, Forest Ecol. Manag., 110, 127–139, 1998.

Bonan, G.B.: Forests and climate change: Forcings, feedbacks, and the climate benefits of forests, Science, 320, 1444–1449, 2008.

Mäkelä, J., Susiluoto, J., Markkanen, T., Aurela, M., Järvinen, H., Mammarella, I., Hagemann, S., and Aalto, T.: Constraining ecosystem model with adaptive Metropolis algorithm using boreal forest site eddy covariance measurements, Nonlin. Processes Geophys., 23, 447-465, doi:10.5194/npg-23-447-2016, 2016.

Rannik, Ü., Peltola, O., and Mammarella, I.: Random uncertainties of flux measurements by the eddy covariance technique, Atmos. Meas. Tech., 9, 5163-5181, doi:10.5194/amt-9-5163-2016, 2016.

Ruosteenoja, K., Raisanen, J. and Pirinen, P.: Projected changes in thermal seasons and the growing season in Finland, Int. J. Climatol., 31, 1473–1487, 2011.

Schaefer, K., et al.: A model-data comparison of gross primary productivity: Results from the North American Carbon Program site synthesis, J. Geophys. Res., 117, G03010, doi:10.1029/2012JG001960, 2012.

Walker A. P., Beckerman A. P., Gu L., Kattge J., Cernusak L. A., Domingues T. F., Scales J. C., Wohlfahrt G., Wullschleger S. D., Woodward F. I. (2014) The relationship of leaf photosynthetic traits – Vcmax and Jmax - to leaf nitrogen, leaf phosphorus, and specific leaf area: a meta-analysis and modeling study. Ecology and Evolution 4, 3218-3235, doi: 10.1002/ece3.1173.

Wohlfahrt, G. and Galvano, M.: Revisiting the choice of the driving temperature for eddy covariance $CO_2$ flux partitioning, Agric. Forest. Met., 237-238, 135-142, 2017.

---

## Author Comment (AC2) · 28 Feb 2017

The manuscript by Thum et al examines the use of SIF to predict GPP in coniferous forests in southern and northern Finland. The authors implement a SIF module in the JSBACH biosphere model and evaluate seasonal and spatially variability against SIF and GPP measurements at leaf, canopy, and ecosystem scale, with focus on spring and autumn transition seasons. A key innovation is the use of active leaf level fluorescence data to understand the seasonal relationship of photochemistry and fluorescence and evaluate model performance. Although many uncertainties exist in the model simulations and in understanding dependencies on environmental vs biochemical effects, the authors show good correlation of observed and simulated variables, providing some confidence for future testing and evaluation, and paving the path for future efforts to scale between leaf and canopy/ecosystem levels. In general, I found this paper interesting and innovative, but it was hard to read at times, and the objective weren't clearly established making results and discussion hard to follow. I recommend a more careful analysis of satellite observations and some general clarifying throughout, but I expect this to be an important study with a few substantial revisions.

We thank the referee for this encouraging feedback and hope we're able to provide improved version of the manuscript based on these recommendations.

Major Comments

My main concern is biasing of GOME-2 time series by filtering of negative SIF values. These data are part of the noise needed in averaging to produce a smoothly varying signal. Because the noise is fixed (0.5 mW m-2 sr-1 nm-1 as mentioned on P9 L4) and doesn't scale with signal, this technique will remove more points in fallwinter-spring when errors are large compared to signal, leading to positive cold season biases, early spring GPP onset, and underestimated seasonal amplitude. This should explain why observed SIF doesn't reach zero level (P13 L5) and why the authors find an opposite phase an opposite phase relationship of GPP and SIF in spring at FI-Hyy in active data (photochemical yield synchronized with SIF) compared to passive data where SIF precedes GPP. I recommend reanalyzing GOME-2 results with negative values.

We thank the referee for this insight and agree. We have thus redone the analysis with the inclusion of negative values.

The authors show that model GPP is systematically early in coniferous forests compared to ground and satellite data, a finding that is consistent with previous studies of cold limited ecosystems. I was hoping the authors could take better advantage of the multiscale observations and new model capabilities to provide explanations at biochemical and environmental levels, especially since the challenge of understanding the spring transition is listed as a motivation for the study. Some speculation is provided in

the discussion (e.g., frost) but not much detail and no mention in the conclusions. I think this is an important enough result and application of new methods as to warrant further discussion. I would like to see the authors discuss what is needed to improve model representation of the spring transition. What would be the effect of seasonal PSII and thermal dissipation? Growing degree days, cold temperature days, and/or frozen soils?

What important environmental controls are included/missing in the Farquhar model?

We agree with the referee that this is one focus of the manuscript, but it has not received enough focus in the earlier version. We therefore added discussion in this topic, which is shown later. There we propose using temperature related changes to the base rates of the biochemical parameters. Here the SIF observations can be used as a valuable evaluation tool for large scale estimates.

In the end, it would be desirable to have process-based description for the cold acclimation processes to properly describe the seasonal cycle of boreal forests, as this would also enhance our skill to predict changes of the carbon cycle in future climate. However, at the moment there is lack of observations to parameterize a large scale model in this respect. The parameters related to the amount of active reaction centers and sustained non-photochemical quenching are steps to this direction, but they would need parameterization in order to be useful in large scale models.

Additionally, also other processes play part. It has been suggested, that slow recovery of Rubisco has influence in spring recovery (Monson et al., 2005). It might not be possible to include all the factors in models, but combining observations and modelling efforts at different scales will hopefully reveal, which processes are most important to be included.

The spring recovery of the forests to its full summertime capacity is a gradual process (Bergh et al., 1998), that can be tracked with several environmental and biological variables (Thum et al., 2009). Air temperature is quite good proxy to be used (e.g. Thum et al., 2009), but the averaged temperature indices might benefit from inclusion of delay due to night frosts (Thum et al., 2017), that might even reverse spring recovery (Ensminger et al., 2004).

Large-scale observations can be very useful, since earlier studies (e.g. Walther et al., 2016) have shown that the temperature sensitivities differ between different regions. This study is a first step in doing that work with more extensive remote sensing data available soon. However, also more data-based approaches are valuable (e.g. Luus et al., 2017, Walther 2016) as they are increasing our understanding of the carbon cycle.

I am also interested in further elaboration of results in autumn at FI-Hyy, in particular, why F' and photochemical yield are strongly delayed relative to GPP in autumn but synchronized in spring.

In earlier version the MONI-PAM results were not discussed in detail, as they've been published also elsewhere (Porcar-Castell, 2011; Kolari et al., 2014), but of course they haven't been shown in this context and therefore there is reason to further elaborate then. In the earlier version they were

discussed in the discussion, but we moved these points to the Results section and added some more detail, now mentioning photoprotection of the needles and the possible differences in the electron transport rate between observation and simulation.

Minor Comments

Abstract and Conclusions – Mostly a discussion of methods and no mention of new results. I suggest discussing at least one new and interesting result from your study. Something about spring or autumn photosynthesis, or using leaf level measurements with satellite data, or comparing model simulations to active and passive fluorescence data.

We added some results, including two points: i) the ability of observed SIF to capture seasonal cycle of photosynthesis at site scale, ii) the goodness of simulated SIF values vs. observations at regional scale.

Figure 1 - Figure legend is difficult to read and it's not clear from figure or caption what is being plotted in panel (A) – legend appears to suggest fluorescence yield as solid red but text refers to photochemical yield. GPP is not shown in panel (A) as stated in caption – please correct.

The font size in the figure legend was increased to make it easier to read. The text was corrected, so that it states that the fluorescence yield is the modelled yield (shown in solid red in the figure). The caption was corrected to say that GPP is in panel B.

Figure 3 – color scheme is confusing especially with multiple variables on 1 plot. keep observations in black and models in color like in figure 1. Use same line styles for same variables (solid for GPP, dashed for SIF).

We remade the figure like suggested.

P5L10: *an indication of the fraction of electrons in the leaf that follow the ChlF pathway

Corrected.

P6L12: *is used in

Corrected.

P7L23 & P9L2: Confusion about overpass time. Here it is stated as 10:30 am but as 9:30 am in Section 2.3.2. Please clarify or correct. Also clarify what it means for the satellite overpass time to last for 100 minutes.

We sincerely apologize for the confusion and would like to explain how this happened. We briefly introduced the properties of GOME-2 in Sect. 2.3.2, where we added "at the equator" to clarify that the actual overpass time depends on the region under investigation. The wording to express the time

for one revolution might have been inappropriate. We clarified this issue by rephrasing the sentence to: "..., while one revolution takes 100 minutes." In Sect. 2.2.3 we added "local solar time" in order to prevent any misunderstanding.

P10L18: I don't see the simultaneous decrease in observed GPP with F'. GPP is already declining on Day 200 while F' appears steady until Day 280. F' decrease is also much more gradual and doesn't reach its minimum until January.

This is right. We have now corrected this part in the text.

P10L25: please elaborate what is meant here - are you suggesting that in low light conditions of spring, most of the absorbed radiation goes into photochemistry thus reducing that available to fluorescence?

Apologies for having a mistake in the subscripts, making the message of the paragraph very unclear. The important message here was connected to only fluorescence yield, that in spring it is the fluorescence yield that is holding back the increase in SIF.

In low light conditions in general, the photosynthetic yield and fluorescence yield are anti-correlated. Actually, with increasing light levels, the fraction of incoming energy used for fluorescence increases and the fraction used for photosynthesis decreases (van der Tol et al., 2009). This is because when photosystem II absorbs light and primary quinone acceptor of PSII $Q_A$ has accepted an electron, it cannot accept another electron before it has passed on the first electron to the subsequent electron carrier. Thus, the proportion of closed reaction centres lead to a reduction in efficiency of photochemistry and increase in fluorescence (Maxwell and Johnson, 2000). We don't have any data that shows different behavior for this in spring. We are sorry for the confusion, due to the wrong subscripts.

P12L1: quantify "reasonably similar" - within 10% of observations? 5%? Regression is slightly lower on average in model

Indeed, it is a good idea to concentrate on this result more deeply, as it is one of the main results of the study. For now we added uncertainties of the slopes in the table 3 and calculated the averages and standard deviations of the different cases. These are now also mentioned in the abstract.

Note, that due to the different fitting algorithm the slopes might differ slightly from the earlier values. Also, due to the inclusion of negative observed SIF values in the analysis, the slopes are not now systematically lower in the model.

P12L8: FI-Sod has lower correlations than FI-Ken.

Yes, it depends on which correlations you're looking at in Table 2. Here we were trying to refer to correlation between modelled GPP and modelled SIF, which is at least 0.92 for other sites and 0.83 for FI-Ken. To clarify this, we added "with each other" to the text.

P12L10: provide reference for peat effect on drought

We decided to replace word "peat" as "humus" as we consider it to be more appropriate term in this context and added reference.

P13L25: what is the magnitude and direction of the seasonal drift in GOME-2 overpass, and what is the likely influence?

Although there are indeed (minor) seasonal variations in the local solar time of GOME-2 overpasses (slightly earlier during NDJ; 10:15 during winter, 10:45 during summer), we do not expect a dramatic influence, because of the senescent vegetation during this period. After including also negative values (as suggested by referee), we obtain SIF values close to zero as it can be anticipated and removed the concerned sentence accordingly. However, the morning overpass of GOME-2 leads to challenging measurement conditions (inclined solar angles) during the winter (mentioned in P14L1).

Inclined solar angles lead to longer photon path lengths, in which case rotational Raman scattering could fill in solar Fraunhofer lines. This might interfere with the SIF retrieval, which relies on the in-filling of Fraunhofer lines. We included these measurements anyways to be able to present a continuous time series (again: we observe SIF values close to zero during winter).

P14L13: please explain what a static temperature response is, the effect on early GPP, and how this could be corrected in the model

We added the following text here:

"*The photosynthesis of forests is often modelled using constant temperature response for the biochemical model parameters $V_{max}$ and $J_{max}$ throughout the year. However, studies have revealed that this assumption does not hold for ecosystems with strong seasonal cycles, but causes overestimation of $CO_2$ fluxes in transition periods. Kolari et al. (2014) found seasonally varying values for leaf level for those parameters from leaf level observations at FI-Hyy. Ueyama et al. (2016) found seasonally varying biochemical model values at four different black spruce forests in Alaska in a model inversion study at eddy covariance sites. In an earlier study using inversion at boreal coniferous forests (Thum et al., 2008), it was found that three forests at northern boreal zone (FI-Hyy, FI-Sod and FI-Ken) had temporal evolution in the biochemical parameters, but a site located on temperate boreal (Norunda, Sweden) did not.*

*Leaf level studies have used temperature acclimation for the changes of biochemical parameters (Wang et al., 1996). Similar results have been obtained for site level results at FI-Sod, where dark acclimated chlorophyll fluorescence observations have been used in combination with eddy covariance observations to disentangle the effect of changing maximum  photosynthetic capacity (Thum et al., 2017).*

*The changes taking place in the needles of conifer forests in winter are numerous to protect the needles in challenging environmental conditions. E.g. the light harvesting complexes are aggregated (Porcar-Castell, 2011) and the xanthophyll cycle enables photoprotection (Ensminger et al., 2004). Some of these processes can be in future be included in a large scale model, as adding changes to the parameters in the ChlF model discussed below, but as changes in the boreal spring happen at quite fast pace and those can be tracked with several different environmental and biological variables (Thum et al., 2009), for large scale applications a temperature related changing of the biochemical parameters might be next step forward and remotely sensed SIF observations provide a very useful evaluation tool in this context.*"

P15L4: add condition "assuming a homogeneous landscape"

We guess that this was meant for page 16… We made the addition there.

References

Bergh, J., McMurtrie, R. E., and Linder, S.: Climatic factors controlling the productivity of Norway spruce: a model-based analysis, Forest Ecol. Manag., 110, 127–139, 1998.

Ensminger, I., Sveshnikov, D., Campbell, D. A., Funk, C., Jansson, S., Lloyd, J., Shibistova, O., and Öquist, G.: Intermittent low temperatures constrain spring recovery of photosynthesis in boreal Scots pine forests, Global Change Biol., 10, 995–1008, 2004.

Luus, K. A., et al. (2017), Tundra photosynthesis captured by satellite-observed solar-induced chlorophyll fluorescence, Geophys. Res. Lett., 44, doi:10.1002/2016GL070842.

Monson R.K., Sparks, J.P., Rosenstiel, T.N., Scott-Denton, L.E., Huxman, T.E., Harley, P.C., Turnipseed, A.A., Burns, S.P., Backlund, B., and Hu J.: Climatic influences on net ecosystem $CO_2$ exchange during the transition from wintertime carbon source to springtime carbon sink in a high-elevation, subalpine forest, Oecologia, 146, 130–147, 2005.

Norton, A. J., Rayner, P. J., Koffi, E. N., and Scholze, M.: Assimilating solar-induced chlorophyll fluorescence into the terrestrial biosphere model BETHY-SCOPE: Model description and information content, Geosci. Model Dev. Discuss., doi:10.5194/gmd-2017-34, in review, 2017.
Thum, T., Aalto, T., Laurila, T., Aurela, M., Hatakka, J., Lindroth, A., and Vesala, T.: Spring initiation and autumn cessation of boreal coniferous forest $CO_2$ exchange assessed by meteorological and biological variables, Tellus 61B, 701-717, 2009.

Thum, T., Aalto, T., Aurela, M., Laurila, T., and Zaehle, S.: Improving the modeling of the seasonal carbon cycle of the boreal forest with chlorophyll fluorescence measurements, in preparation, 2017.

Ueyama, M., Tahara, N., Iwata, H., Euskirchen, E.S., Ikawa, H., Koyayashi, , Nagano, H., Nakai, T., and Harazono, Y.: Optimization of a biochemical model with eddy covariance measurements in

black spruce forests of Alaska for estimating $CO_2$ fertilization effects, Agr. Forest Meteorol., 222, 98-111, 2016.

Wang, K. Y., Kellomäki, S., and Laitinen, K.: Acclimation of photosynthetic parameters in Scots pine after three-year exposure to elevated $CO_2$ and temperature, Agr. Forest Meteorol., 82, 195–217, 1996.

---

## Author Comment (AC3) · 28 Feb 2017

[revised manuscript text omitted]

In spring all studied ChlF variables increased simultaneously with photosynthesis. However, in autumn, the decline in $\Phi_P$ was much slower than in F'. This slow decline might be due to dark autumns, as the needles do not suffer from excess light levels and thus the downregulation of the light harvesting machinery can be much lower (Kolari et al., 2014). The yield of photochemistry and fluorescence declines during winter because the yield of NPQ increased in a process regulated by air temperature (Porcar-Castell, 2011). Therefore, the needles will be more protected against high light levels during the spring when the air temperature remains low and photosynthesis is curbed.

The photosynthesis of forests is often modelled using constant temperature response for the biochemical model parameters $V_{max}$ and $J_{max}$ throughout the year. However, studies have revealed that this assumption does not hold for ecosystems with strong seasonal cycles, but causes overestimation of $CO_2$ fluxes in transition periods, at least in spring. Kolari et al. (2014) found seasonally varying values for leaf level for those parameters from leaf level observations at FI-Hyy. Ueyama et al. (2016) found seasonally varying biochemical model values at four different black spruce forests in Alaska in a model inversion study at eddy covariance sites. In an earlier study using inversion at boreal coniferous forests (Thum et al., 2008), it was found that three forests at northern boreal zone (FI-Hyy, FI-Sod and FI-Ken) had temporal evolution in the biochemical parameters, but a site located on temperate boreal (Norunda, Sweden) did not.

Leaf level studies have used temperature acclimation for the changes of biochemical parameters (Wang et al., 1996). Similar results have been obtained for site level results at FI-Sod, where dark acclimated chlorophyll fluorescence observations have been used in combination with eddy covariance observations to disentangle the effect of changing maximum photosynthetic capacity (Thum et al., 2017).

The changes taking place in the needles of conifer forests in winter are numerous to protect the needles in challenging environmental conditions. E.g. the light harvesting complexes are aggregated (Porcar-Castell, 2011) and the xanthophyll cycle enables photoprotection (Ensminger et al., 2004). Some of these processes can be in future be included in a large scale model, as adding changes to the parameters in the ChlF model discussed below, but as changes in the boreal spring happen at quite fast pace and those can be tracked with several different environmental and biological variables (Thum et al., 2009), for large scale applications a temperature related changing of the biochemical parameters might be next step forward and remotely sensed SIF observations provide a very useful evaluation tool in this context."

In addition, tThe number of active PSII reaction centers (parameter $q_{Ls}$ in the chlorophyll fluorescence model) has been shown to change seasonally in boreal environments (Porcar-Castell, 2011). However, in our implementation we assumed it to be a constant 0.5, as there is no theory to predict variations of this parameter at larger scales. Similarly the rate constant of sustained thermal dissipation (parameter $k_{NPQs}$ in the chlorophyll fluorescence model) incorporates seasonal variation in boreal forests (Porcar-Castell, 2011), but for the same reasons it was kept as zero in our model runs. The comparison with the data nevertheless suggests that these assumptions are justified at the time and spatial scales of our analysis.

However, since the seasonal cycle was captured quite well by the model at FI-Hyy, even though the seasonally variable parameters that control yield were not considered, some concerns connected with the model are evident. The link to the Farquhar model causes the simulated ChlF variables to have a pronounced seasonal cycle similar to the measurements, even though the light reactions of the SIF model do not include the seasonal changes that take place in the leaves. While this could be considered as a counterargument against our approach, the fact that we can generate an appropriate time series with the environmental controls of the Farquhar model suggests that our approach maybe a sensible choice when attempting to simulate SIF at ecosystem and larger scales.

**4.2 Satellite data**

The satellite SIF observations have a clear-sky bias, which may affect the seasonality of these data. Furthermore, illumination and viewing geometries affect the observed SIF (Joiner et al., 2013). The high wintertime values observed by GOME-2 for SIF at the Finnish sites were not likely connected to photosynthetic activity and they were also below the error threshold of the observations. It is likely that this would be more visible in the northern sites, where SIF values are generally lower. During winter there are some warm days when the forests might be active, although the activity is not as comparable in magnitude to the summer time as these values would suggest. The phenomenon is not connected to snow reflectance as SIF measures emissions only from the green component of the canopy. Similar behavior has been observed at some other boreal forest regions in GOME 2 data, but it is not common in the boreal region (Walther et al., 2016). Moreover, this was also noticed in the SCIAMACHY and OCO 2 data (Supplementary Material, B). These characteristics would indicate that care must be taken when SIF data is used for evaluation of modelled photosynthesis or in data assimilation in boreal regions. 
[revised manuscript text omitted]

[Figure]

[Figure]

Fig. 1

[Figure]

Fig. 2

[Figure]

[Figure]

Fig. 3

[Figure]

[Figure]

Fig. 4

[Figure]

[Figure]

Fig. 5

[Figure]

[Figure]

Fig. 6

[Figure]

[Figure]

Fig. 7

---

## Author Comment (AC4) · 28 Feb 2017

**S1 Figures from site level simulations**

[Figure]

Fig. S1. (a) Five years of incoming photosynthetically active radiation (PAR) (blue line), modelled quantum yield of fluorescence ($\Phi_f$)(black line) and sun-induced chlorophyll fluorescence (SIF) (red line) and (b) modelled fraction of absorbed photosynthetic active radiation by vegetation (fAPAR) (black line) and leaf area index (LAI) (red line). The daily values have been smoothed by a moving 10-day window.

[Figure]

Fig. S2. (a) The diurnal cycles of observed prevailing fluorescence signal as measured with PAM fluorometry (F') and quantum yield of photochemistry in PSII ($\Phi$p) from MONI-PAM and modelled quantum yield of fluorescence ($\Phi$f ) and sun-induced chlorophyll fluorescence (SIF) as averaged from 12 sunny days during the growing season, (b) the gross primary production (GPP) as measured from the flux tower (blue line) and GPP simulated by JSBACH (red line), (c) photosynthetically active radiation (PAR) observation from the flux tower (blue line) and from the MONI-PAM observation place in the canopy (black line).

[Figure]

Fig. S3. (a) The diurnal cycles of observed prevailing fluorescence signal as measured with PAM fluorometry (F') and quantum yield of photochemistry in PSII (Φp) from MONI-PAM and modelled quantum yield of fluorescence (Φf) and sun-induced chlorophyll fluorescence (SIF) as averaged from 12 cloudy days during the growing season, (b) the gross primary production (GPP) as measured from the flux tower (blue line) and GPP simulated by JSBACH (red line), (c) photosynthetically active radiation (PAR) observation from the flux tower (blue line) and from the MONI-PAM observation place in the canopy (black line).